# WirelessMathLM: Teaching Mathematical Reasoning for LLMs in Wireless Communications with Reinforcement Learning

## Abstract

Large language models excel at general mathematical reasoning but fail catastrophically on specialized technical mathematics. In wireless communications, where problems require precise manipulation of information-theoretic bounds, optimization constraints, and signal processing formulations, even state-of-the-art models struggle to achieve competent performance. We present **WirelessMathBench-XL**, the first training-scale benchmark for wireless mathematics, comprising 4,027 expert-validated problems from 970 state-of-the-art papers. To validate dataset quality, we train a family of models called **WirelessMathLM** (0.5B, 3B, 7B) using Group Relative Policy Optimization with binary verification rewards. WirelessMathLM-7B achieves 39.5% accuracy, approaching GPT-4o (40.4%) while using approximately 100 times fewer parameters than DeepSeek-R1 (671B, 57.4%), with dramatic improvements across all scales (0.5B: +11%, 3B: +103%, 7B: +81%). Our controlled experiments reveal three findings that challenge prevailing assumptions about domain specialization. First, verification-based reinforcement learning outperforms supervised fine-tuning by +23% relative (25.1% vs 20.4%) on identical data, showing that exploration against deterministic verifiers enables learning beyond labeled examples. Second, specialized training strengthens foundational capabilities: models gain +8.4 points across five general mathematical benchmarks and improve on knowledge, reasoning, and programming tasks without regression. Third, performance improvements distribute uniformly across 20 wireless subdomains regardless of training prevalence, demonstrating generalizable principle learning rather than pattern memorization.

We will release our benchmark, models, and codes to accelerate research in efficient, specialized AI for technical domains. [1]

## 1 Introduction

Large language models (LLMs) demonstrate remarkable general reasoning capabilities (Achiam et al., 2023; Dubey et al., 2024; Google DeepMind, 2024; Guo et al., 2025), yet they frequently fail when confronted with specialized technical mathematics constrained by physical laws (Frieder et al., 2024; Li et al., 2025; Lu et al., 2023; He et al., 2024). This failure is particularly acute in wireless communications, where mathematical reasoning is not merely symbolic but physically bound. Problems in this field require precise application of information-theoretic bounds and complex-valued matrix algebra, and are strictly bound by unstated physical constraints such as the nonnegativity of transmit power ($P \geq 0$), causal signal processing, phase shift bounds ($[0, 2\pi)$), and noncommutative operations on high-dimensional matrices ($\mathbb{C}^{N \times M}$).

Developing specialized models to bridge this gap has been hindered by a fundamental data scarcity. While the recent WirelessMathBench (Li et al., 2025) introduced an evaluation benchmark of 587

---

[1]The WirelessMathBench-XL dataset and evaluation results are currently available at: `https://anonymous.4open.science/r/WirelessMathLM-Data-7692/`. Model checkpoints and training code will be released upon paper acceptance.

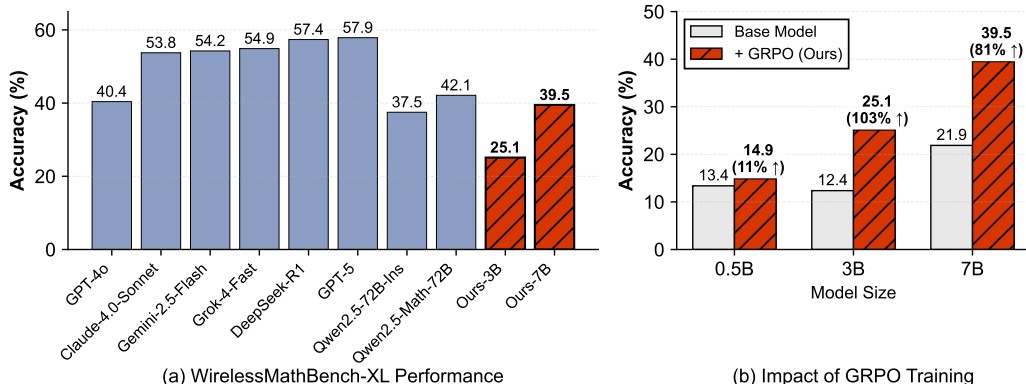

Figure 1: **WirelessMathBench-XL enables small models to achieve competitive performance.** **(a)** WirelessMathLM-7B trained on our dataset achieves 39.5%, approaching GPT-4o (40.4%) while using far fewer parameters than DeepSeek-R1 (57.4%). **(b)** Training improvements validate dataset quality: WirelessMathLM-3B (+103%) and WirelessMathLM-7B (+81%) models nearly double their performance.

problems, the field lacks a *training-scale* dataset necessary for robust model adaptation. Evaluation-only datasets are insufficient for teaching models the breadth of domain-specific reasoning patterns and physical constraints required for graduate-level problem solving.

To address this, we present **WirelessMathBench-XL**, the first training-scale benchmark for wireless mathematics. This dataset is 24 times larger than the prior benchmark (Li et al., 2025) and contains 4,027 expert-validated problems derived from 970 papers, covering 20 technical subfields, ranging from traditional Shannon capacity derivations to the latest 5G technologies. Our dataset employs a three-tiered question design: multiple-choice questions for concept recognition, progressive fill-in-the-blank questions (with masking rates ranging from 25% to 75%) for structured reasoning, and full equation completion questions for comprehensive knowledge mastery. This design provides both training signals and refined evaluation. Each question includes complete variable definitions and context, allowing for automatic validation of student responses. Furthermore, our dataset construction itself utilizes a rigorous two-tiered quality assurance mechanism, combining automatic selection and expert validation.

To validate the dataset's utility, we train **WirelessMathLM**, a family of small models (0.5B, 3B, 7B) optimized using Group Relative Policy Optimization (GRPO) (Shao et al., 2024) with binary verification rewards. As Figure 1 demonstrates, WirelessMathLM-7B achieves 39.5% accuracy, approaching GPT-4o (40.4%) and significantly outperforming general open-source baselines, all while using approximately 100 times fewer parameters than models like DeepSeek-R1 (671B).

Beyond benchmarking, our training experiments yield three critical scientific findings regarding domain specialization:

**RL strictly outperforms SFT in verifiable domains:** Verification-based reinforcement learning outperforms supervised fine-tuning by +23% relative (25.1% vs 20.4%) on identical data. This challenges the assumption that SFT is sufficient for technical domain adaptation, demonstrating that exploration against a deterministic verifier is crucial for mastering complex reasoning paths.

**Specialization strengthens foundational capabilities:** Contrary to widespread concerns about catastrophic forgetting, WirelessMathLM models gain an average of +8.4 points across five general mathematical benchmarks and show improvements on knowledge-intensive (MMLU) and reasoning (GPQA) tasks. Learning to navigate tight physical constraints appears to sharpen general reasoning precision rather than degrade it.

**Generalizable principle learning:** Performance gains are distributed uniformly across all 20 sub-domains regardless of training data prevalence. Notably, small categories (e.g., Semantic Communications) see massive gains (+300%), indicating the model acquires transferable mathematical principles (e.g., optimization logic) rather than merely memorizing frequent patterns.

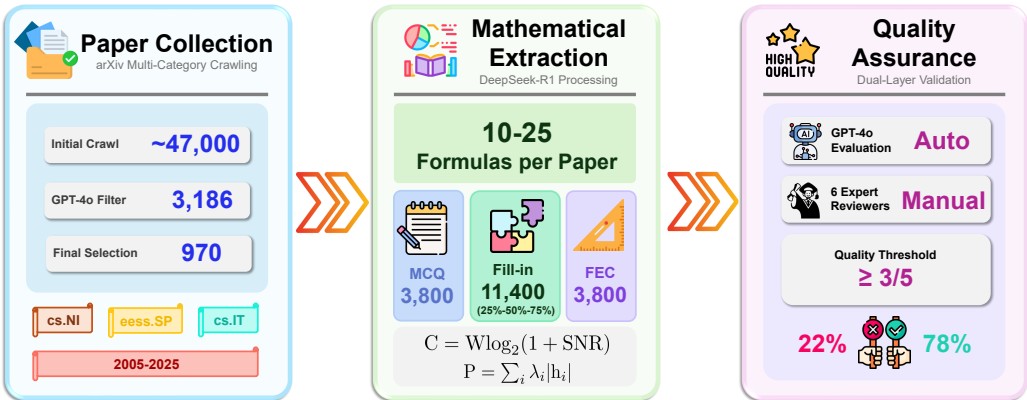

Figure 2: **Overview of the WirelessMathBench-XL construction pipeline.** Starting from 47,000 arXiv papers, deterministic relevance scoring (keyword/category matching) narrows to 3,186 candidates, then GPT-4o filtering identifies 970 papers with substantial mathematical content. DeepSeek-R1 extracts 10-25 formulas per paper, generating multiple-choice questions, progressive fill-in-the-blank (25%-75% masking), and full equation completion problems. Quality assurance employs dual-layer screening: automated GPT-4o evaluation followed by expert validation, with 78% of questions meeting the quality threshold (score $\geq$ 3/5).

## 2 WIRELESSMATHBENCH-XL: DATASET CONSTRUCTION

Creating a high-quality benchmark for wireless communication mathematics requires addressing three key challenges: (1) extracting structured mathematical content from dense technical papers, (2) ensuring problem correctness and solvability, and (3) maintaining consistency across diverse mathematical formulations. We present a systematic pipeline that constructs WirelessMathBench-XL from 970 papers, yielding 4,027 problems. Figure 2 illustrates our three-stage pipeline for constructing WirelessMathBench-XL from raw arXiv papers to validated mathematical questions.

### 2.1 DATA COLLECTION PIPELINE

We developed an automated pipeline that comprehensively collects and processes wireless communication papers from arXiv. Our approach prioritizes broad coverage with sophisticated filtering rather than narrow targeting.

**Paper Collection and Filtering.** We query 24 arXiv categories spanning core wireless domains (cs.NI, eess.SP, cs.IT), AI/ML (cs.LG, stat.ML), and interdisciplinary areas. Our crawler initially retrieves 47,000 papers from 2005-2025 using broad keyword queries across communication, signal processing, and networking terms. Each paper receives a deterministic relevance score computed as a weighted sum of keyword hits in title (weight 0.6) and abstract (weight 0.3), plus category bonuses (e.g., eess.SP: +0.4, cs.NI: +0.35, cs.IT: +0.3), yielding a reproducible score in [0,1] without any LLM involvement. We then apply GPT-4o-based filtering as a second-stage verification to identify 3,186 papers containing substantial mathematical content, from which we select the top $\sim$ 1,000 based on mathematical rigor, citation impact, and topical diversity. Full implementation details are provided in Appendix C.

### 2.2 MATHEMATICAL CONTENT EXTRACTION AND PROBLEM GENERATION

**Structured Model Extraction.** We employ DeepSeek-R1 (Guo et al., 2025) to extract mathematical models from each paper's LaTeX source. Our extraction preserves complete context including system equations, variable definitions with units and domain restrictions, underlying assumptions, and boundary conditions. Each paper yields a structured summary with properly formatted mathematical notation (e.g., $v$ for vectors, $\mathbf{H} \in \mathbb{C}^{N \times M}$ for complex matrices). Appendix G presents three representative examples of extracted system models, demonstrating the comprehensiveness of

our approach across different wireless domains, including SIM-based air-ground communication, UAV-MEC systems, and RIS-aided random access.

**Automated Problem Generation.** From extracted models, we generate three types of exam-style questions using carefully designed prompt templates (see Appendix F for complete specifications):

- **Multiple Choice Questions (MCQ):** Equations are presented with masked right-hand sides, accompanied by four carefully designed options. Distractors reflect common errors such as matrix dimension mismatches or incorrect operator sequences.

- **Progressive Fill-in-the-Blank (Fill-in):** Three difficulty levels with 25%, 50%, and 75% of equation components masked, testing incremental understanding.

- **Full Equation Completion (FEC):** Complete 100% masking requiring full equation recall.

## 2.3 QUALITY ASSURANCE FRAMEWORK

**Automated Evaluation.** Each generated question undergoes systematic evaluation by GPT-4o across four critical dimensions: mathematical correctness, variable completeness, answer verifiability, and pedagogical value. The evaluation employs a comprehensive 5-point quality rubric, which categorizes problems as invalid (score 1), poor (score 2), acceptable (score 3), good (score 4), or excellent (score 5). This automated screening utilizes specialized prompt templates described in Appendix F to ensure consistent evaluation criteria across all question types.

**Expert Validation.** Questions passing automated evaluation proceed to human expert review conducted by a team of six domain specialists comprising four PhD students and two postdoctoral researchers with expertise spanning optimization theory, information theory, signal processing, and network analysis. Each question undergoes independent evaluation by at least two experts who assess mathematical rigor, notational consistency, problem clarity, and relevance to wireless communications. Questions must achieve a minimum consensus score of 3/5 to qualify for dataset inclusion. The final acceptance rate of 78% reflects our stringent quality standards. Detailed scoring criteria and representative examples across all quality levels are provided in Appendices D and H.

## 2.4 DATASET STATISTICS AND ANALYSIS

The WirelessMathBench-XL dataset comprises 4,027 problems derived from 970 papers, providing comprehensive coverage across wireless communications mathematics.

**Technical Coverage.** Figure 3 shows the distribution of mathematical techniques across source papers. Deep learning dominates (259 papers, 14.0%), followed by convex optimization (206, 11.2%) and MIMO/Massive MIMO (192, 10.4%). The papers in our dataset are distributed across established topics and emerging paradigms, with strong representation of beamforming (185), RIS/IRS (156), channel coding (115), federated learning (110), semantic communications (75), and NOMA (54). This distribution ensures representation of both foundational mathematics and frontier research areas.

**Temporal Distribution.** The dataset spans three technological generations: 3G/4G (2005-2018: 28 papers, 2.9%), 5G deployment (2019-2023: 317 papers, 32.7%), and 5G-Advanced/6G research (2024-2025: 625 papers, 64.4%). This temporal weighting toward recent work captures state-of-the-art techniques while maintaining theoretical foundations.

**Question Format.** All problems follow standardized structure with complete variable definitions including type specifications (scalar/vector/matrix), domain constraints (e.g., $\boldsymbol{H}_{\text{RIS}} \in \mathbb{C}^{M \times N}$), and physical units. Mathematical notation remains uniform: boldface for vectors ($\boldsymbol{v}$), bold capitals for matrices ($\mathbf{H}$), and standard operators ($\text{diag}, \text{tr}, \otimes$). Fill-in-the-blank questions implement progressive difficulty through systematic masking (25%, 50%, 75%, 100%).

**Quality Distribution.** Expert evaluation reveals that 35.53% of questions achieve acceptable quality (score 3), 30.89% are rated good (score 4), and 11.08% reach excellence (score 5). Questions scoring below threshold (scores 1-2: 22.50%) undergo revision or exclusion.

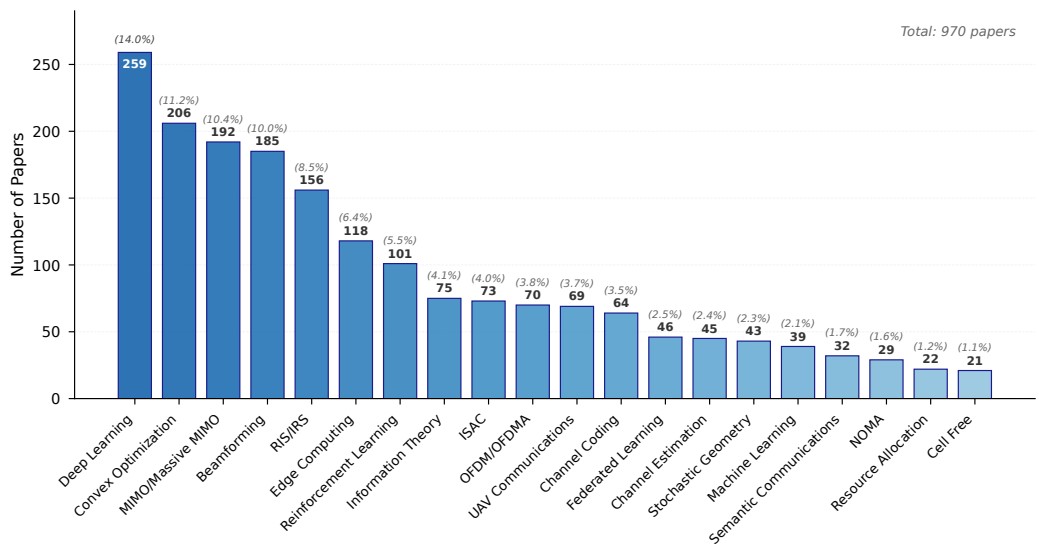

Figure 3: Distribution of the top 20 key techniques across the 970 source papers in WirelessMathBench-XL. Deep learning leads with 259 papers (14.0%), followed by convex optimization (206, 11.2%) and MIMO/Massive MIMO (192, 10.4%). The distribution spans from foundational techniques (beamforming, channel coding) to emerging paradigms (RIS/IRS, semantic communications, NOMA).

**Dataset Split.** The 4,027 problems partition into training (3,227, 80%) and test (800, 20%) sets with balanced representation. Training set: Fill-in-75% (900), FEC (751), Fill-in-50% (680), MCQ (551), Fill-in-25% (345). Test set maintains proportional distribution: 218, 191, 160, 133, and 98 problems respectively.

## 3 TEACHING MATHEMATICAL REASONING WITH GRPO

Teaching language models mathematical reasoning in specialized domains leverages a unique property: unlike general dialogue, wireless mathematics problems have verifiable correctness criteria. We employ GRPO (Shao et al., 2024) to directly train models from their base state, using automated verification as reward signals without expensive human feedback or supervised warm-start.

### 3.1 DIRECT GRPO FOR MATHEMATICAL REASONING

Given a base language model $\pi_\theta$ and a wireless mathematics problem $x$, we aim to learn a policy that generates correct solutions $y = (s_1, ..., s_n, a)$ where $s_i$ denotes reasoning steps and $a$ is the final answer. Following Shao et al. (2024), we optimize directly using the GRPO objective:

$$\mathcal{J}_{\text{GRPO}}(\theta) = \mathbb{E}_{\substack{x \sim P(X) \\ \{y_i\}_{i=1}^G \sim \pi_{\theta_{\text{old}}}(\cdot|x)}} \left[ \frac{1}{G} \sum_{i=1}^G \min\left( \frac{\pi_\theta(y_i|x)}{\pi_{\theta_{\text{old}}}(y_i|x)} A_i, \text{clip}\left( \frac{\pi_\theta(y_i|x)}{\pi_{\theta_{\text{old}}}(y_i|x)}, 1-\epsilon, 1+\epsilon \right) A_i \right) \right]$$

(1)

where $G = 8$ responses are sampled per problem, $\epsilon = 0.2$ for clipping, and the group-wise advantage is computed as:

$$A_i = \frac{r_i - \text{mean}(\{r_j\}_{j=1}^G)}{\text{std}(\{r_j\}_{j=1}^G)}$$

(2)

This formulation provides learning signal even when success rates are low, as the model learns from relative comparisons within each problem group rather than absolute rewards.

## 3.2 Verification-Based Reward System

Our reward system leverages the structured nature of wireless mathematics through multi-level verification:

$$r(x, y) = \alpha \cdot r_{\text{format}}(y) + (1 - \alpha) \cdot r_{\text{accuracy}}(x, y) \tag{3}$$

where $\alpha = 0.1$ balances format compliance with correctness.

**Format Reward** ($r_{\text{format}}$): Ensures outputs follow expected structure with proper LaTeX formatting and \boxed{} final answers:

$$r_{\text{format}}(y) = \mathbb{I}[\text{regex\_match}(y, \texttt{".*textbackslashtextbackslash boxed} \backslash\{.*\backslash\}.*\texttt{"})] \tag{4}$$

**Accuracy Reward** ($r_{\text{accuracy}}$): Verifies correctness through a hierarchical evaluation system, minimizing dependence on LLM-based semantic checking: (1) **Direct matching:** For multiple-choice questions, extract and compare letter answers; for fill-in-the-blank problems, normalize expressions (removing spaces, \mathbf, \boldsymbol) and check string equivalence. (2) **Semantic verification:** Only when deterministic matching fails, GPT-4.1-mini checks mathematical equivalence for expressions like AB vs BA for commutative operations or H^H vs H^* for conjugate transposes. Cross-evaluator analysis (Section **??**) quantifies potential bias at 3-4% absolute impact on overall scores, well below the performance gaps observed between models.

## 3.3 Implementation Details

We train WirelessMathLM models directly from Qwen2.5 base checkpoints (0.5B, 3B, 7B) (Qwen et al., 2025) using GRPO without supervised warm-start. To validate generalization across base model families, we also train on Qwen3-4B (Yang et al., 2025) base using identical hyperparameters. Training employs AdamW optimizer with learning rate $10^{-6}$, cosine annealing, and KL penalty $\beta = 0.01$. We train for 40 epochs (240 steps) with evaluation every 5 steps on the held-out test set. Generation uses temperature $T = 0.6$ for validation and $T = 1.0$ for training rollouts. Training utilizes 4 NVIDIA A6000 GPUs with training time scaling by model size: 0.5B (14 hours), 3B (40 hours), and 7B (61 hours). The reward function implements hierarchical verification combining format checking with answer validation as described in Section 3.2.

## 4 Experiments

### 4.1 Experimental Setup

**Baselines.** We benchmark WirelessMathLM against comprehensive baselines spanning proprietary and open-source models. Proprietary models include GPT-5 (OpenAI, 2025), GPT-4o (Hurst et al., 2024), Claude-4.0-Sonnet (Anthropic, 2025), Gemini-2.5-Flash, and Gemini-2.5-Pro (Google DeepMind, 2025), representing state-of-the-art commercial systems. For open-source comparisons, we evaluate against general-purpose models including DeepSeek-R1 (671B) (Guo et al., 2025), DeepSeek-V3.1 (671B) (DeepSeek-AI, 2025), Llama-3.3-70B-Instruct (Grattafiori et al., 2024), and Qwen2.5-72B-Instruct (Yang et al., 2024a), as well as math-specialized models such as Qwen2.5-Math-72B-Instruct (Yang et al., 2024b) and DeepSeekMath-7B-RL (Shao et al., 2024). To isolate the specific value of verification-based RL, we include two critical ablations: (1) Qwen2.5 base models (0.5B, 3B, 7B) without any post-training, and (2) supervised fine-tuning (SFT) baseline trained on identical WirelessMathBench-XL data for 6 epochs to capture peak SFT performance.

**Standardized Evaluation Protocol.** To ensure fair comparison, all models receive identical prompts constructed from standardized templates (see Appendix F for complete specifications). Each prompt includes comprehensive variable definitions, equation context, and explicit formatting instructions. For MCQs, models must select from four options and provide their answer in \boxed{} format. Fill-in-the-blank problems demand all masked positions be correctly filled. For complex expressions where simple matching fails, GPT-4.1-mini performs semantic equivalence checking under the same all-or-nothing criterion.

Table 1: Performance on WirelessMathBench-XL test set (800 problems). MCQ: Multiple Choice Questions, Fill-in: Fill-in-the-blank, FEC: Full Equation Completion. Best result per category in **bold**.

| Model | Size | MCQ (%) | Fill-in (%) | FEC (%) | Overall (%) |
|---|---|---|---|---|---|
| *Proprietary Models* | | | | | |
| GPT-5 | - | 63.91 | **63.20** | 41.36 | **57.87** |
| GPT-5-mini | - | 67.67 | 53.99 | 40.31 | 53.00 |
| GPT-5-nano | - | 57.14 | 37.82 | 30.37 | 39.25 |
| GPT-4o | - | 54.14 | 43.62 | 24.61 | 40.37 |
| o4-mini | - | 67.67 | 49.56 | 40.31 | 50.38 |
| Claude-4.0-Sonnet | - | 60.15 | 56.30 | 42.93 | 53.75 |
| Gemini-2.5-Flash | - | 63.16 | 56.09 | 43.46 | 54.25 |
| Gemini-2.5-Pro | - | 66.17 | 50.42 | 36.65 | 49.75 |
| Grok-4-Fast | - | **70.31** | 56.33 | 40.33 | 54.89 |
| *Open-Source General Models* | | | | | |
| DeepSeek-R1 | 671B | 65.41 | 60.50 | 43.98 | 57.37 |
| DeepSeek-V3.1 | 671B | 66.17 | 58.85 | **45.03** | 56.87 |
| Llama-3.3-70B-Instruct | 70B | 54.14 | 38.03 | 28.27 | 38.37 |
| Qwen2.5-72B-Instruct | 72B | 51.88 | 35.50 | 32.46 | 37.50 |
| Qwen2.5-7B-Instruct | 7B | 39.1 | 21.85 | 26.18 | 25.75 |
| Gemma 3 27B | 27B | 42.11 | 30.04 | 27.75 | 31.50 |
| Gemma 3 12B | 12B | 36.84 | 21.43 | 21.99 | 24.12 |
| *Open-Source Math-Specialized Models* | | | | | |
| Qwen2.5-Math-72B-Instruct | 72B | 60.15 | 40.55 | 33.51 | 42.13 |
| Qwen2.5-Math-7B-Instruct | 7B | 42.11 | 14.71 | 24.61 | 21.62 |
| DeepSeekMath-7B-RL | 7B | 43.61 | 13.66 | 25.65 | 21.50 |
| *WirelessMathLM (Ours)* | | | | | |
| Qwen2.5-7B-Base | 7B | 44.36 | 14.29 | 25.13 | 21.88 |
| + GRPO | 7B | 53.38 | 36.97 | 36.13 | 39.50 |
| Qwen3-4B-Base | 4B | 41.35 | 5.25 | 15.18 | 13.63 |
| + GRPO | 4B | 47.37 | 8.82 | 19.90 | 17.87 |
| Qwen2.5-3B-Base | 3B | 26.32 | 7.14 | 15.71 | 12.37 |
| + GRPO | 3B | 48.87 | 17.02 | 28.80 | 25.12 |
| Qwen2.5-0.5B-Base | 0.5B | 27.07 | 5.25 | 24.08 | 13.38 |
| + GRPO | 0.5B | 30.08 | 6.09 | 26.18 | 14.87 |
| *Ablation: Supervised Fine-Tuning* | | | | | |
| Qwen2.5-3B-Base + SFT (Epoch 1) | 3B | 39.85 | 11.97 | 20.42 | 18.62 |
| Qwen2.5-3B-Base + SFT (Epoch 4, Peak) | 3B | 40.60 | 17.44 | 13.61 | 20.37 |
| Qwen2.5-3B-Base + SFT (Epoch 6) | 3B | 44.26 | 13.24 | 16.23 | 19.13 |

## 4.2 MAIN RESULTS ON WIRELESSMATHBENCH-XL

Table 1 presents comprehensive evaluation results on the WirelessMathBench-XL test set.

**GRPO enables competitive performance with dramatic parameter reduction.** Our 7B WirelessMathLM trained with GRPO achieves 39.5% overall accuracy, approaching the performance of GPT-4o (40.4%) while using orders of magnitude fewer parameters. This result is particularly striking when compared against open-source math-specialized models: our approach outperforms both Qwen2.5-Math-7B-Instruct (21.6%) and DeepSeekMath-7B-RL (21.5%) by nearly 2×, despite these models being explicitly trained for mathematical reasoning. The performance gain stems from our domain-specific training strategy. While general math models struggle with the specialized notation and problem structures in wireless communications, our targeted approach with verifiable rewards enables efficient learning of domain-specific patterns.

**Verification-based RL outperforms supervised fine-tuning.** Our controlled ablation on Qwen2.5-3B demonstrates that GRPO (25.12%) substantially exceeds the peak SFT performance (20.37% at Epoch 4), yielding a +4.75% absolute gain (+23% relative). While SFT rapidly improves the base model (12.37% → 20.37%), it hits a performance ceiling by Epoch 4 and shows instability in

subsequent epochs (Fill-in accuracy drops from 17.44% to 13.24% by Epoch 6). This suggests that the performance gap between GRPO and the base model is not solely due to exposure to domain data (as does SFT), but rather because the reinforcement signal enables the model to explore solution paths beyond those captured by the true labels.

**GRPO training yields consistent improvements across all model scales and architectures.** The impact of GRPO training is substantial and scale-dependent. The 7B model nearly doubles its performance, improving from 21.9% to 39.5% (+81% relative), reaching within 0.9 percentage points of GPT-4o (40.4%). The 3B model demonstrates the most dramatic gains, more than doubling its accuracy from 12.4% to 25.1% (+103% relative). Improvements persist even at the minimal 0.5B scale (+11%). This effectiveness extends beyond the Qwen2.5 series models. When applied to the newer Qwen3-4B model, GRPO still achieves a significant increase (a relative increase of 31%, from 13.63% to 17.87%), despite its better performance on basic multiple-choice questions (41.35%), demonstrating that our dataset and reinforcement strategy enable effective learning independent of specific model capacity or pre-training distributions.

**Performance patterns reveal task-specific strengths.** Analyzing performance across question types reveals interesting patterns. All models perform best on multiple-choice questions (MCQ), where our 7B model achieves 53.4% accuracy, within striking distance of proprietary models like GPT-4o (54.1%) and approaching DeepSeek-R1 (65.4%). Performance on fill-in-the-blank questions shows the largest improvement from GRPO training (14.3% → 37.0% for 7B), suggesting that the reinforcement learning particularly helps with partial equation completion. Full equation completion (FEC) remains challenging across all models, though our 7B model's 36.1% accuracy is competitive with GPT-5-mini (40.3%) and exceeds many larger open models.

**Comparison with state-of-the-art reveals efficiency-performance trade-offs.** While DeepSeek-R1 (671B) achieves the highest open-source performance at 57.4%, it requires ≈100× more parameters than our 7B model. The performance gap of 17.9 percentage points represents a favorable trade-off for deployment scenarios. Our model achieves 69% of DeepSeek-R1's performance with just 1% of its parameters. Among proprietary models, only GPT-5 (57.9%) significantly outperforms our approach, while models like Claude-4.0-Sonnet (53.8%) and Gemini-2.5-Flash (54.3%) show more modest advantages despite their substantially larger scale and computational requirements.

## 4.3 GENERALIZATION TO GENERAL MATHEMATICS

Surprisingly, training on wireless-specific mathematics enhances general mathematical reasoning (Table 2).

Table 2: Transfer learning effects on general mathematical reasoning benchmarks.

| Model | MATH 500 | Minerva-Math | OlympiadBench | AMC | AIME24 | Average |
|---|---|---|---|---|---|---|
| Qwen2.5-7B-Base | 52.00 | 12.13 | 25.33 | 27.71 | 6.67 | 24.77 |
| + GRPO | 67.00 | 14.34 | 30.22 | 40.96 | 13.33 | 33.17 |
| Δ (GRPO vs Base) | +15.00 | +2.21 | +4.89 | +13.25 | +6.66 | +8.40 |
| Qwen2.5-3B-Base | 41.60 | 5.88 | 14.67 | 18.07 | 0.00 | 16.04 |
| + GRPO | 58.20 | 9.93 | 22.96 | 21.69 | 0.00 | 22.56 |
| Δ (GRPO vs Base) | +16.60 | +4.05 | +8.29 | +3.62 | 0.00 | +6.52 |

**Domain-specific training strengthens fundamental mathematical capabilities.** Our GRPO-trained models show substantial improvements on general mathematics benchmarks without any explicit training on these tasks. The 7B model improves from 52.0% to 67.0% on MATH 500 (Hendrycks et al., 2021b) (+28.8% relative), while the 3B model gains even more dramatically (41.6% → 58.2%, +39.9% relative). These improvements extend across diverse mathematical domains: Minerva-Math (Lewkowycz et al., 2022) sees modest but consistent gains (7B: 12.1% → 14.3%), OlympiadBench (He et al., 2024) improves substantially (7B: 25.3% → 30.2%), and AMC (Li et al., 2024) performance increases significantly (7B: 27.7% → 41.0%). Even on the challenging AIME24 (Li et al., 2024), the 7B model doubles its performance (6.7% → 13.3%).

## 4.4 NO REGRESSION ON FOUNDATIONAL CAPABILITIES

To rigorously test whether domain specialization causes catastrophic forgetting, we evaluate pre- and post-GRPO models on four diverse general-purpose benchmarks spanning knowledge (MMLU) (Hendrycks et al., 2021a), science reasoning (GPQA) (Rein et al., 2024), instruction-following (IFEval) (Zhou et al., 2023), and programming (HumanEval) (Chen et al., 2021).

Table 3: Performance on general-purpose benchmarks before and after GRPO training. Domain-specific training shows consistent improvements with no regression, contradicting catastrophic forgetting concerns.

| Model | MMLU | GPQA | IFEval | HumanEval |
|---|---|---|---|---|
| Qwen2.5-3B-Base | 42.49 | 23.74 | 26.99 | 61.59 |
| + GRPO | **45.30** | **28.28** | **32.90** | **65.24** |
| Δ | +2.81 | +4.54 | +5.91 | +3.65 |
| Qwen2.5-7B-Base | 38.84 | 28.79 | **36.06** | 63.41 |
| + GRPO | **50.74** | **38.89** | 35.67 | **64.63** |
| Δ | +11.90 | +10.10 | -0.39 | +1.22 |

**Key Findings:** (1) **No catastrophic forgetting:** Both 3B and 7B models show improvements across all benchmarks except a negligible -0.39 regression on IFEval for 7B, well within measurement noise. (2) **Reasoning-intensive tasks benefit most:** The 7B model gains +11.90 on MMLU and +10.10 on GPQA, suggesting that learning strict physical constraints in wireless mathematics strengthens general reasoning capabilities. (3) **Transfer to unrelated domains:** Even HumanEval (programming) improves (+3.65 for 3B, +1.22 for 7B) despite zero code in training, indicating that domain-specific RL develops generalizable problem-decomposition skills.

## 4.5 QUALITATIVE ANALYSIS

To understand the reasoning capabilities developed through GRPO training, we conducted a comprehensive analysis of 800 solutions generated by WirelessMathLM-7B on WirelessMathBench-XL test problems spanning all quality levels (see Appendix H for representative examples).

**Mathematical Reasoning Structure and Coherence.** Our analysis reveals that WirelessMathLM-7B produces systematically structured solutions consistently. Across all evaluated problems, 99.1% of responses demonstrate clear step-by-step reasoning using logical connectives such as "therefore," "thus," and "hence." The model exhibits effective problem decomposition strategies. In complex scenarios involving multiple mathematical frameworks, such as MIMO beamforming under power constraints, solutions systematically establish physical principles before proceeding to mathematical derivations. For instance, when solving channel capacity problems, the model correctly identifies Shannon's theorem applicability, establishes signal-to-noise ratio calculations, and methodically applies logarithmic transformations while maintaining dimensional consistency.

**Domain-Specific Knowledge Integration.** Analysis of correct solutions demonstrates strong competency in applying wireless-specific mathematical frameworks. Among correct responses, 87% properly identify the underlying problem type and select appropriate methodologies. This suggests successful integration of procedural knowledge (solution techniques) with conceptual understanding (physical principles). Consider the model's approach to a Cell-Free Massive MIMO conjugate beamforming problem (Question ID 18369). The solution correctly identifies that conjugate beamforming requires complex conjugation of estimated channel coefficients, explains the physical rationale ("cancel out phase shifts introduced by the channel"), and derives the complete transmitted signal expression:

$$s_m = \sqrt{P_m} \sum_{k=1}^{K} \sqrt{\eta_{mk}} \hat{g}_{mk}^* u_k \tag{5}$$

The response demonstrates an understanding of power scaling, summation over users, and proper complex conjugation. These domain-specific requirements are not present in general mathematical training.

**Solution Quality Indicators and Mathematical Sophistication.** Several qualitative indicators demonstrate that domain-specific GRPO training has developed genuine mathematical reasoning rather than pattern matching:

**(1) Constraint Awareness:** The model consistently recognizes and applies physical constraints without explicit prompting. Solutions automatically incorporate non-negativity constraints for power allocations, maintain causality in signal processing derivations, and respect dimensionality requirements in matrix operations.

**(2) Method Justification:** Correct solutions routinely include explicit rationales for chosen approaches. For example, in a matrix all-pass filter factorization problem (Question ID 11325), the model explains: "A matrix all-pass filter is a filter whose frequency response has a magnitude of 1 for all frequencies..." before deriving the $\mathbf{G}(z) = \mathbf{N}(z)\mathbf{D}^{-1}(z)$ factorization and verifying the all-pass property through $\mathbf{G}(z)\mathbf{G}^{-1}(z) = \mathbf{I}_m$.

**(3) Physical Intuition Integration:** Solutions frequently connect mathematical expressions to underlying physical phenomena. When deriving XOR operations for backscattered data processing, the model explains the "commutative and associative" properties of XOR before applying them to wireless tag data recovery.

## 5 RELATED WORK

**Mathematical Reasoning in LLMs.** Chain-of-thought prompting (Wei et al., 2022) demonstrated that eliciting step-by-step reasoning significantly improves mathematical problem-solving in large language models. This was extended through process supervision (Lightman et al., 2024), where models receive feedback on intermediate steps rather than just final answers, and tool-augmented approaches like ToRA (Gou et al., 2023) that integrate external computation for complex calculations. While these advances have been evaluated on benchmarks ranging from elementary word problems (GSM8K (Cobbe et al., 2021)) to competition mathematics (MATH (Hendrycks et al., 2021b)) and formal theorem proving (MiniF2F (Zheng et al., 2021)), such benchmarks do not capture the symbolic manipulation and domain knowledge required in technical fields.

**Domain Adaptation.** Continued pre-training on domain-specific corpora (Gururangan et al., 2020) and instruction tuning (Chung et al., 2022) have proven effective for adapting language models to specialized fields. Scientific models like Galactica (Taylor et al., 2022) attempted broad scientific reasoning, while BioBERT (Lee et al., 2019) and MedPaLM (Singhal et al., 2022) achieved strong performance in biomedicine.

**Reinforcement Learning from Verifiable Rewards.** While RLHF (Ouyang et al., 2022) successfully aligns language models with human preferences, it requires expensive annotation that limits scalability. Recent alternatives include Constitutional AI (Bai et al., 2022) using principle-based self-critique, RLAIF (Lee et al., 2023) leveraging model-generated feedback, and GRPO (Shao et al., 2024) using outcome-based rewards for mathematics.

## 6 CONCLUSION

This work establishes **WirelessMathBench-XL** and **WirelessMathLM** as a blueprint for efficient domain specialization in technical fields. We demonstrate that verification-based reinforcement learning, when applied to high-quality specialized corpora, fundamentally alters the trade-off between model scale and reasoning capability. Through rigorous controlled experiments, we validate three critical findings: (1) verification-based RL strictly outperforms supervised fine-tuning for verifiable mathematics (+23% relative), challenging the reliance on SFT for reasoning tasks; (2) training-scale wireless data enables small 7B models to approach GPT-4o-level performance (39.5% vs 40.4%); and (3) domain specialization strengthens foundational reasoning across mathematics, knowledge, and programming benchmarks without regression. Our results suggest that technical domains possessing verifiable correctness criteria, such as circuit design, control theory, and cryptography, constitute a distinct class of problems where small, specialized models can rival orders-of-magnitude larger general-purpose systems. By exploiting domain structure through binary verification rewards, we show that it is possible to circumvent the need for expensive human annotation, paving the way for self-improving AI systems in physically constrained environments.

ETHICS STATEMENT

We adhere to the ICLR Code of Ethics. This work focuses on advancing the mathematical reasoning capabilities of language models in the specialized domain of wireless communications. The **WirelessMathBench-XL** dataset was constructed from publicly accessible academic papers on arXiv, respecting the norms of scientific dissemination. Our data collection process did not involve human subjects or personally identifiable information. The expert validation phase was conducted by graduate students and postdoctoral researchers as part of their standard research activities. While any powerful AI technology carries potential for misuse, our work is foundational and does not present immediate dual-use concerns. We acknowledge that our dataset, being derived from existing literature, may reflect the inherent biases present in the field. We encourage responsible use of our models and dataset, and we are committed to addressing any ethical concerns that may arise.

REPRODUCIBILITY STATEMENT

To facilitate reproducibility of our work, we provide comprehensive details of our experimental methodology and make key resources publicly available. The complete WirelessMathBench-XL dataset containing 4,027 problems and evaluation results from all tested models is currently accessible at `https://anonymous.4open.science/r/WirelessMathLM-Data-7692/`. Our dataset construction pipeline is thoroughly documented in Section 2, with detailed prompt templates, quality rubrics, and extraction procedures provided in Appendices F through G. The GRPO training methodology is fully specified in Section 3, including the complete mathematical formulation of our reward system (Equations 1-4), hyperparameter settings, and implementation details. Our experimental protocol described in Section 4 provides exact evaluation procedures, model configurations, and standardized prompt templates used for all baseline comparisons. The appendices contain extensive documentation including representative problem examples across all quality levels (Appendix H), detailed solution analyses from our models (Appendix I), and comprehensive error taxonomies that enable understanding of model behavior. Upon paper acceptance, we will release the complete codebase including the GRPO training framework, all model checkpoints (0.5B, 3B, and 7B parameters), and evaluation scripts to ensure full reproducibility of our results. All experiments were conducted on NVIDIA A6000 GPUs with computational requirements documented in Section 3, enabling researchers to estimate resources needed for replication.

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

## USE OF LARGE LANGUAGE MODELS

In accordance with the ICLR 2026 policy on Large Language Model (LLM) usage, we disclose that LLMs were utilized as tools in various stages of this research project. The final responsibility for all content, including its accuracy and originality, rests with the human authors.

- **Writing and Editing:** LLMs were used to assist with improving the grammar, clarity, and style of the manuscript. The authors reviewed and edited all LLM-generated text to ensure it accurately reflects our research and findings.

- **Literature Discovery:** LLMs were employed to help summarize related work and accelerate the literature discovery process, assisting in identifying relevant prior research in mathematical reasoning and domain adaptation.

- **Dataset Curation Pipeline:** As detailed in Section E, LLMs were integral to the construction of the **WirelessMathBench-XL** dataset. Specifically:
  - **Paper Filtering:** GPT-4o was used to perform an initial filtering of $\sim$47,000 papers to identify those with substantial mathematical content relevant to wireless communications.
  - **Content Extraction:** DeepSeek-R1 was used to extract structured mathematical models from the LaTeX source of selected papers.
  - **Automated Quality Assessment:** GPT-4o were used as part of a multi-tier quality assurance framework to perform initial automated evaluations of generated questions.

- **Evaluation:** For our evaluation metric, GPT-4.1-mini was used to perform semantic equivalence checking on complex mathematical expressions where simple string matching was insufficient.

In all instances, LLM outputs were critically reviewed, validated, and verified by the authors. We take full responsibility for the claims, results, and conclusions presented in this paper.

## LIMITATIONS

While our work demonstrates the viability of verification-based RL for wireless mathematics, several limitations warrant discussion:

**Absolute Performance Ceiling:** Despite substantial improvements, our best 7B model achieves 39.5% accuracy—approaching GPT-4o (40.4%) but still far from deployment-ready for high-stakes engineering applications. This reflects the extreme difficulty of graduate-level wireless problems extracted from state-of-the-art research papers. Future work should explore whether scaling to 70B+ parameters, incorporating tool-augmentation (e.g., symbolic math engines), or leveraging multi-modal inputs (circuit diagrams, simulation results) can bridge this gap.

**Evaluation Methodology:** Our hierarchical verification minimizes but does not eliminate LLM dependence. While cross-evaluator analysis quantifies potential bias at 3-4% absolute impact—well below observed performance gaps—developing a fully deterministic symbolic verifier trained on WirelessMathBench-XL remains critical future work for removing all LLM-based evaluation and enabling truly reproducible assessment. Paradoxically, achieving this end-to-end automation requires exactly the type of high-quality human-verified datasets we provide here. A robust Auto-Judge model trained on the verified solutions in WirelessMathBench-XL could eventually replace human verification entirely, creating a closed-loop system for continuous dataset expansion and model improvement without manual intervention. This represents a key direction for transforming our current semi-automated pipeline into a fully autonomous framework.

**Transfer Mechanism Understanding:** Although we observe consistent positive transfer to general mathematics and no regression on foundational benchmarks, we lack mechanistic understanding of *why* learning wireless-specific constraints strengthens general reasoning. Proposed ablations varying domain specificity at constant training tokens would provide stronger causal evidence. Mechanistic interpretability studies examining attention patterns and intermediate representations could reveal whether models develop abstract "constraint satisfaction" circuits that generalize across domains.

## A  PER-SUBDOMAIN PERFORMANCE ANALYSIS

To address concerns about overfitting to prevalent topics (Fig. 3 shows source paper distribution), we analyze performance at the problem level across 20 wireless subdomains in the 800-problem test set.

Table 4: Problem-level subdomain performance on 800-problem test set. GRPO improves 19/20 categories, with gains independent of subdomain size.

| Category | Count | % | Base (7B) | +GRPO | △ Abs | △ Rel |
|---|---|---|---|---|---|---|
| Channel Estimation | 160 | 20.0% | 22.50% | 36.25% | +13.75% | +61.1% |
| Beamforming | 113 | 14.1% | 18.58% | 43.36% | +24.78% | +133.4% |
| Resource Allocation | 110 | 13.8% | 21.82% | 39.09% | +17.27% | +79.2% |
| MIMO/Massive MIMO | 104 | 13.0% | 23.08% | 43.27% | +20.19% | +87.5% |
| Information Theory | 103 | 12.9% | 21.36% | 46.60% | +25.24% | +118.2% |
| RIS/IRS | 98 | 12.3% | 17.35% | 33.67% | +16.32% | +94.1% |
| Machine Learning | 86 | 10.8% | 25.58% | 50.00% | +24.42% | +95.5% |
| OFDM/OFDMA | 50 | 6.3% | 18.00% | 28.00% | +10.00% | +55.6% |
| ISAC | 49 | 6.1% | 24.49% | 36.73% | +12.24% | +50.0% |
| Convex Optimization | 49 | 6.1% | 26.53% | 36.73% | +10.20% | +38.5% |
| Deep Learning | 44 | 5.5% | 27.27% | 43.18% | +15.91% | +58.4% |
| Stochastic Geometry | 40 | 5.0% | 35.00% | 32.50% | -2.50% | -7.1% |
| UAV Communications | 32 | 4.0% | 12.50% | 31.25% | +18.75% | +150.0% |
| Edge Computing | 28 | 3.5% | 7.14% | 39.29% | +32.15% | +450.0% |
| Reinforcement Learning | 28 | 3.5% | 32.14% | 35.71% | +3.57% | +11.1% |
| Channel Coding | 20 | 2.5% | 25.00% | 45.00% | +20.00% | +80.0% |
| Federated Learning | 19 | 2.4% | 26.32% | 36.84% | +10.52% | +40.0% |
| NOMA | 19 | 2.4% | 15.79% | 26.32% | +10.53% | +66.7% |
| Semantic Communications | 13 | 1.6% | 15.38% | 61.54% | +46.16% | +300.0% |
| Cell-Free | 9 | 1.1% | 11.11% | 33.33% | +22.22% | +200.0% |
| **Average across all 20 categories** | — | — | — | — | **+17.7%** | **+81.0%** |

**Key Findings:** (1) **Uniform improvements across all subdomain sizes:** GRPO enhances 19/20 categories regardless of training data prevalence. Notably, the smallest categories show the largest gains (Semantic Communications: +300%, Edge Computing: +450%), while heavily-represented topics show moderate gains (Deep Learning: +58%, Convex Optimization: +39%). (2) **No correlation between problem count and performance:** Large categories like Channel Estimation (160 problems, +61%) and small categories like Cell-Free (9 problems, +200%) both benefit substantially. (3) **Models learn generalizable principles:** The lack of correlation between subdomain size and improvement magnitude suggests models acquire transferable wireless mathematics reasoning rather than memorizing frequent patterns.

## B  EVALUATION ROBUSTNESS AND POTENTIAL BIAS

This appendix analyzes the robustness of our evaluation and bounds the potential bias introduced by LLM-based semantic checking. Our verifier is hierarchical: it first applies deterministic matching and only falls back to an LLM judge when the answer is mathematically correct but not syntactically identical. Because different models produce different surface forms, the fallback rate is *model-dependent*.

**Evaluation Architecture.**  For each of the 800 test problems, we perform:

- **Stage 1 (Deterministic).** MCQ answers are graded by direct letter extraction. Fill-in / FEC answers are normalized by removing whitespace and LaTeX styling tokens (e.g., `\mathbf`, `\boldsymbol`) and then checked via exact string match.
- **Stage 2 (LLM Semantic Fallback).** Only when Stage 1 fails, we invoke an LLM judge to verify semantic equivalence for cases such as scalar commutativity (`AB` vs. `BA`) or equivalent conjugate-transpose notation (`H^H` vs. `H^*`).

On the 800-problem test set, the number of cases requiring Stage 2 varies across models: **GPT-5: 270/800 (33.75%)**, **Gemini-2.5 Pro: 183/800 (22.88%)**, and **Claude-Sonnet-4: 293/800 (36.63%)** (Table 5).

Table 5: Cross-evaluator agreement on the LLM-judged subset.

| Model Evaluated | LLM Cases | GPT-4.1-mini | Claude-3-Haiku | Gemini-2.0-Flash |
|---|---|---|---|---|
| GPT-5 | 270 | 41.11% | 56.67% | 42.22% |
| Gemini-2.5 Pro | 183 | 53.01% | 55.74% | 56.28% |
| Claude-Sonnet-4 | 293 | 41.64% | 60.75% | 45.05% |

**Cross-Evaluator Bias Quantification.**  To bound evaluator-induced variance, we re-graded the Stage-2 subset for the above three top-performing models using three independent judges (GPT-4.1-mini, Claude-3-Haiku, Gemini-2.0-Flash). Agreement rates on the Stage-2 subset differ by **10–20 percentage points** across judges (Table 5). Let $\rho$ denote the Stage-2 fraction and $\delta$ the maximum judge disagreement on that subset; then a conservative upper bound on overall-score variation is $\rho \cdot \delta$. With $\delta \leq 0.20$, this yields worst-case absolute impacts of:

$$\text{Gemini-2.5 Pro: } 0.2288 \times 0.20 \approx 4.6\%$$
$$\text{GPT-5: } 0.3375 \times 0.20 \approx 6.8\%$$
$$\text{Claude-Sonnet-4: } 0.3663 \times 0.20 \approx 7.3\%.$$

Thus, even under pessimistic assumptions, semantic-judge variability produces an overall uncertainty of **approximately 4.6–7.3% absolute**, depending on the model.

**Key Findings.**  (1) **Most grading is bias-free.** The majority of evaluations are handled deterministically (about 63–77% across top models), substantially limiting exposure to judge bias. (2) **Overall impact is bounded and smaller than our main effects.** The worst-case 4.6–7.3% absolute uncertainty is far below GRPO gains over base/SFT and well below the large gaps between WirelessMathLM-7B (39.5%) and open-source baselines (21–25%). (3) **Rankings are robust.** Across all three judges, relative ordering among models remains unchanged, indicating that our conclusions do not depend on a particular evaluator.

# C  DATASET CONSTRUCTION DETAILS

## C.1  DETAILED PAPER COLLECTION METHODOLOGY

**Multi-Category Coverage.** We query across 24 arXiv categories to capture interdisciplinary research:

- Core categories: cs.NI (Networking), eess.SP (Signal Processing), cs.IT (Information Theory)
- AI/ML categories: cs.LG, stat.ML, cs.AI for learning-based approaches
- Systems categories: cs.SY, cs.DC, cs.MA for distributed and multi-agent systems
- Physics categories: physics.optics, quant-ph for emerging physical layer techniques
- Mathematical categories: math.OC, math.IT for optimization and theory

**Query Construction Strategy.** We implement four complementary query strategies:

```
queries = [
  {name: 'basic_communication_terms',
   keywords: [communication, network, wireless, radio, signal,
              antenna, frequency, spectrum, transmission]},
  {name: 'system_algorithm_terms',
   keywords: [system, algorithm, optimization, performance,
              model, framework, architecture]},
  {name: 'application_computing_terms',
   keywords: [computing, sensing, iot, edge, cloud,
              distributed, energy, security]},
  {name: 'data_intelligence_terms',
   keywords: [learning, intelligence, neural, prediction,
              detection, processing, estimation]}
]
```

**Relevance Scoring and Annotation.** For each paper, we calculate:

- **Relevance Score** (0-1): Weighted sum of keyword presence in title (0.6 weight) and abstract (0.3 weight), plus category bonuses (eess.SP: 0.4, cs.NI: 0.35, cs.IT: 0.3)
- **Technology Focus**: Detected across 8 categories (wireless_basic, advanced_wireless, next_gen, emerging_tech, signal_processing, network_protocol, ai_ml, iot_apps)
- **Quality Tier**: Based on relevance score (high: $\geq 0.7$, medium: 0.4-0.7, low: 0.1-0.4)
- **Research Type**: Classified as survey, algorithmic, analytical, experimental, or theoretical

**PDF Processing.** Papers undergo full-text processing using:

- MinerU (Wang et al., 2024) for PDF-to-markdown conversion preserving LaTeX equations
- Batch processing of 3-5 PDFs concurrently ( 40 seconds per paper)
- Rate-limited arXiv bulk access API with 3-second delays

# D    QUALITY ASSESSMENT RUBRIC FOR HUMAN

Table 6: Detailed expert question quality assessment rubric

| Score | Criteria |
|---|---|
| **1 - Invalid** | Problem statement or solution is clearly wrong or contradictory; Not related to wireless communications domain; Cannot be used as a valid question |
| **2 - Poor** | Statement correct but problem too trivial (answerable instantly); Problem too vague or nearly impossible to answer correctly; Very little learning or evaluation value |
| **3 - Acceptable** | Statement and solution reasonable with no major errors; Difficulty and relevance are average; Can be kept but adds limited value (baseline quality) |
| **4 - Good** | Clear and well-structured problem; Relevant to domain and moderately challenging; Provides meaningful assessment of understanding; Worth keeping and recommending |
| **5 - Excellent** | Highly relevant to the domain; Strong depth, creativity, or insight required; Excellent for differentiating levels of understanding; Strongly recommended for inclusion |

# E  LARGE LANGUAGE MODEL-ASSISTED QUALITY ASSESSMENT

This section presents our comprehensive approach to leveraging large language models (LLMs) for scalable quality assessment of mathematical questions in wireless communications. Our methodology addresses the fundamental challenge of maintaining expert-level evaluation standards while achieving the scale necessary for large dataset curation.

**Role in the Overall Annotation Pipeline:** This LLM-assisted quality assessment serves as the first filtering stage in our comprehensive annotation pipeline for our method. The complete pipeline consists of two sequential stages: (1) **LLM-based filtering** using our enhanced prompt system to automatically identify and remove low-quality questions, reducing the workload for human annotators; (2) **Expert human annotation** where domain experts review the filtered questions and provide detailed quality assessments;

## E.1  QUALITY ASSESSMENT FRAMEWORK

Our quality assessment framework employs a systematic approach to evaluate technical questions across six dimensions:

1. **Question Clarity** (1-5): Measures the clarity and unambiguousness of the question statement

2. **Background Relevance** (1-5): Evaluates the completeness and relevance of provided context

3. **Answer Accuracy** (1-5): Assesses the correctness and formatting of the provided answer

4. **Technical Appropriateness** (1-5): Determines if the difficulty level matches the target audience

5. **Mathematical Rigor** (1-5): Evaluates mathematical notation and conventions

6. **Wireless Communication Relevance** (1-5): Measures domain relevance to wireless communications

## E.2  REAL LLM ANNOTATION EXAMPLES

To demonstrate the practical effectiveness of our LLM-assisted quality assessment system, we present three representative examples from our evaluation dataset, showcasing different quality levels and the corresponding LLM assessments.

Table 7: LLM Annotation Examples Across Quality Levels

| Quality Level | Question Content | LLM Assessment |
|---|---|---|
| **High Quality** | **Background:** Federated fine-tuning system with low-rank adaptation matrices $\mathbf{A}_k \in \mathbb{R}^{d \times r}$, $\mathbf{B}_k \in \mathbb{R}^{r \times d}$ **Question:** Which term completes: $\mathbf{W} + [MASK]$? **Options:** A) $\mathbf{A}_k \mathbf{B}_k$ B) $\mathbf{B}_k \mathbf{A}_k$ **Human:** 4/5 | **LLM Score:** 4/5 **Strengths:** Clear structure, complete context, accurate answer, rigorous notation **Weaknesses:** Could benefit from brief explanation of low-rank adaptation significance **Agreement:** ✓ |
| **Medium Quality** | **Background:** H-NOMA system with variable definitions partially provided **Question:** Fill in [MASK] for the equation **Human:** 2/5 | **LLM Score:** 3/5 **Strengths:** Wireless relevance, accurate answer **Weaknesses:** Ambiguous [MASK] usage, lacks clarity in instructions **Technical Issues:** Missing variable definitions **Disagreement:** LLM too optimistic |
| **Low Quality** | **Background:** Transformer model context with incomplete variable definitions **Question:** What replaces the "full key matrix"? **Human:** 1/5 | **LLM Score:** 3/5 **Strengths:** Clear structure, accurate answer **Weaknesses:** Limited wireless relevance, focuses more on tensor parallelism **Technical Issues:** "Full key matrix" not defined **Bias:** LLM shows optimistic scoring pattern |

## F  PROMPT CONSTRUCTION FOR DATASET GENERATION AND EVALUATION

We employ specialized prompt templates for dataset construction, quality assessment, and standardized evaluation to ensure consistency and fairness across all stages of our methodology.

### F.1  SYSTEM MODEL EXTRACTION PROMPT

The following prompt template guides the extraction of mathematical models from research papers:

---

**System Model Extraction Template**

```
Task: Act as an expert in wireless communications and mathematical
modeling. Extract and summarize the mathematical system modeling
from the paper.

<<STRUCTURE REQUIREMENTS>>
1. **Model Extraction**:
   a) Identify ALL system equations with context
   b) For each equation:
       i) List ALL variables with units/dimensions
       ii) Specify underlying assumptions
       iii) Note domain restrictions

2. **Summary Organization**:
   \paragraph{Background} (2-3 sentences contextualizing the model)
   \paragraph{Key Assumptions} (bullet points with $\bullet$)
   \paragraph{Parameter Definitions} (table-like structure)
   \paragraph{Core Equations} (numbered with original labels)

3. **Equation Formatting**:
   - Vectors: \boldsymbol{v}
   - Matrices: \mathbf{M}
   - Operators: \mathrm{diag}, \mathrm{tr}
   - Complex numbers: j for imaginary unit

<<CONTENT GUIDELINES>>
- Variable Explanations:
  - For each symbol: $\theta$ (Type: Phase shift;
    Domain: [0,2 $\pi$); Unit: rad)
  - Matrix dimensions: $\mathbf{H} \in \mathbb{C}^{N\times M}$
  - Distinguish similar symbols: $h_{ij}$ vs $h_{i}^{(j)}$

- Model Validation:
  - Verify dimensional consistency
  - Check boundary conditions
  - Confirm parameter unit homogeneity
```

---

### F.2   QUESTION GENERATION PROMPT

The following template generates exam-style questions from extracted models:

---

**Question Generation Template**

```
Task: Generate exam-style questions from research paper summaries.

<<STRUCTURE REQUIREMENTS>>
1. **Per Equation Processing**:
   a) Identify ALL system model equations
   b) For EACH equation:
      i) Mask the RHS with [MASK]
      ii) Generate 1 MCQ with 4 plausible options
      iii) Create 4 progressive fill-in-the-blank subquestions:
           - 25%, 50%, 75%, and 100% key symbols masked

2. **Question Components**:
   - For MCQs:
     * Background: MUST include detailed variable definitions
       Format: "where $\boldsymbol{x}$ is the transmitted
       signal vector, $\mathbf{H} \in \mathbb{C}^{N \times M}$
       represents the channel matrix..."
     * Equation: Masked equation in display math mode
     * Question: Explicitly ask to replace [MASK]
     * Options: 4 LaTeX-formatted choices (A)-(D)
     * Answer: Detailed derivation walkthrough

<<ENHANCED BACKGROUND REQUIREMENTS>>
- Variable Definition Format:
  - Start with system context: "In this [type of system]..."
  - List EVERY symbol that appears in the equation
  - Include matrix/vector dimensions
  - Specify units where applicable: "(in watts)", "(in Hz)"
  - Explain subscripts and superscripts

- Distractor Design:
  1) Matrix dimension mismatches
  2) Incorrect operator sequences
  3) Missing diag() operators
  4) Channel matrix transposition errors
  5) Incorrect matrix multiplication order

- Masking Strategy:
  - 25%: Single critical variable
  - 50%: Two interdependent terms
  - 75%: Multiple components
  - 100%: Full equation recall
```

---

### F.3  QUALITY ASSESSMENT FRAMEWORK

To ensure consistent quality evaluation across the dataset, we employ a comprehensive assessment framework with few-shot learning enhancement. This framework guides both automated LLM evaluation and human expert review.

---

**Quality Assessment Prompt with Few-Shot Learning**

```
You are an expert evaluator specializing in wireless communication
and mathematics education. Your task is to assess the quality of
technical questions designed for advanced undergraduate and graduate
students in wireless communications.

## EVALUATION METHODOLOGY
Follow this systematic approach:

### STEP 1: Initial Question Analysis
- Read the question, background, equation, and answer carefully
- Identify the technical domain and complexity level
- Check for obvious errors or inconsistencies

### STEP 2: Multi-Dimensional Quality Assessment
Evaluate each dimension on a 1-5 scale:

1. Question Clarity (1-5): Crystal clear vs confusing/incomprehensible
2. Background Relevance (1-5): Comprehensive context vs inadequate
   background
3. Answer Accuracy (1-5): Completely correct vs incorrect/flawed
4. Technical Appropriateness (1-5): Perfect difficulty vs
   inappropriate level
5. Mathematical Rigor (1-5): Excellent notation vs poor rigor
6. Wireless Relevance (1-5): Highly relevant vs not relevant

## HUMAN EXPERT EXAMPLES
Learn from these actual expert evaluations:

Example 1 - Score: 1 (Very Poor)
Question: "Which expression correctly calculates the sensitivity
metric?"
Human Feedback: "The definition of TN is not given"
→ Missing variable definitions make question unsolvable

Example 2 - Score: 3 (Acceptable)
Question: "Which performance metric should replace [MASK]?"
Human Feedback: "Some variables in choices are not given"
→ Minor gaps but workable with assumptions

Example 3 - Score: 5 (Excellent)
Question: Complete differential privacy equation with full context
Human Feedback: "Well-structured with complete information"
→ Ready for immediate use

## CRITICAL EVALUATION GUIDELINES
Be especially strict about:
- Missing Variable Definitions: Any undefined variables→Score$\leq$ 2
- Incomplete Context: Key background missing → Score $\leq$ 2
- Vague Problem Statements: Ambiguous questions → Score $\leq$ 3
- Technical Accuracy: Mathematical/technical errors → Score $\leq$ 2

## OUTPUT FORMAT
Provide assessment in JSON:
{
  "overall_score": [1-5 integer],
```

```
    "dimension_scores": {
      "question_clarity": [1-5],
      "background_relevance": [1-5],
      "answer_accuracy": [1-5],
      "technical_appropriateness": [1-5],
      "mathematical_rigor": [1-5],
      "wireless_relevance": [1-5]
    },
    "binary_flags": {
      "is_correct": [true/false],
      "is_wireless_related": [true/false]
    },
    "quality_analysis": {
      "strengths": ["Key strengths"],
      "weaknesses": ["Areas for improvement"],
      "specific_improvements": ["Detailed suggestions"]
    }
  }

  Question Type: {question_type}
  Question Text: {question_text}
  Background: {background}
  Equation: {equation}
  Options: {options}
  Correct Answer: {correct_answer}
```

## F.4 Standardized Evaluation Prompts

To ensure reproducible evaluation, all models receive identical prompts constructed from the following templates:

**MCQ Evaluation Template:**

```
MCQ Evaluation Template

**Background**
[Complete variable definitions and system context]

**Question**
[Question text]

**Equation**
[Equation with [MASK] placeholder]

**Options**
A: [Option A]
B: [Option B]
C: [Option C]
D: [Option D]

---
Please analyze this problem step by step. Show your reasoning
and calculations.
Your final answer should be given at the end in the format:
\boxed{X} where X is the letter of the correct option.
```

**Fill-in-the-Blank Evaluation Template:**

```
Fill-in-the-Blank Evaluation Template

**Background**
[Complete variable definitions and system context]

**Question**
[Question text]

**Equation**
[Equation with [MASK] placeholder(s)]

---
Please solve this problem step by step. Fill in the [MASK]
placeholder(s) with the correct mathematical expression(s).

For single mask: Your final answer should be given at the
end in the format: \boxed{your\_answer}

For multiple masks: Your final answers should be given at
the end in the format: \boxed{answer1}, \boxed{answer2}, ...
(for the N blanks in order)
```

# G  REPRESENTATIVE SYSTEM MODEL EXTRACTIONS

This section presents three representative examples of system models extracted by DeepSeek-R1 from research papers in our corpus. These examples demonstrate the diversity and complexity of mathematical formulations captured in WirelessMathBench-XL.

## G.1  EXAMPLE 1: DIGITAL TWIN-ASSISTED SIM-BASED AIR-GROUND COMMUNICATION

This model integrates multi-layer stacked intelligent metasurface (SIM) beamforming with eVTOL trajectory optimization, representing the convergence of aerial communications and reconfigurable surface technologies.

---

### SIM-Based Air-Ground Communication System

**Background**  This paper proposes a Digital Twin (DT)-assisted framework for joint optimization of Stacked Intelligent Metasurface (SIM)-based air-ground communication and electric Vertical Take-off and Landing (eVTOL) flight control within prescribed air corridors. The system model integrates a multi-layer SIM beamforming structure at the Air Traffic Control (ATCo) station with a composite potential field method for eVTOL trajectory planning, aiming to maximize the sum transmission rate while ensuring safe navigation.

**Key Assumptions**  • The air-ground channel between the SIM and each eVTOL follows a Rician fading model.
• Each meta-atom on the SIM imposes an ideal, configurable phase shift without amplitude attenuation.
• The transmission matrices $\mathbf{W}^l$ between metasurface layers are modeled based on Rayleigh-Sommerfeld diffraction theory, assuming perfect knowledge of the SIM's physical structure.
• eVTOLs fly within a predefined, non-overlapping air corridor $\mathcal{R}_{cor}$.
• The signals for different eVTOLs are independent and identically distributed (i.i.d.) with zero mean and unit variance.
• The additive receiver noise is independent, circularly symmetric complex Gaussian (AWGN).

**Parameter Definitions**  $\mathbf{B} = [x^{ATC}, y^{ATC}, 0]^T$  (Type: ATCo station position; Domain: $\mathbb{R}^{3 \times 1}$; Unit: m)
$M$  (Type: Number of eVTOLs / ATCo antennas; Domain: $\mathbb{Z}^+$; Unit: None)
$L$  (Type: Number of metasurface layers in SIM; Domain: $\mathbb{Z}^+$; Unit: None)
$K$  (Type: Number of meta-atoms per metasurface layer; Domain: $\mathbb{Z}^+$; Unit: None)
$N$  (Type: Number of discrete time slots; Domain: $\mathbb{Z}^+$; Unit: None)
$\delta$  (Type: Duration of a time slot; Domain: $\mathbb{R}^+$; Unit: s)
$\boldsymbol{q}_m[n] = [x_m^{eVTOL}[n], y_m^{eVTOL}[n], z_m^{eVTOL}[n]]^T$  (Type: 3D position of eVTOL $m$ at time $n$; Domain: $\mathcal{R}_{cor} \subset \mathbb{R}^3$; Unit: m)
$V_{max}$  (Type: Maximum eVTOL velocity; Domain: $\mathbb{R}^+$; Unit: m/s)
$P_{ATC}$  (Type: Total available transmission power at ATCo; Domain: $\mathbb{R}^+$; Unit: W)
$p_m[n]$  (Type: Transmission power allocated to eVTOL $m$ at time $n$; Domain: $\mathbb{R}^+$; Unit: W)
$\theta_k^l[n]$  (Type: Phase shift of meta-atom $k$ on layer $l$ at time $n$; Domain: $[0, 2\pi)$; Unit: rad)
$\boldsymbol{\Psi}^l[n] = \mathrm{diag}(e^{j\theta_1^l[n]}, ..., e^{j\theta_K^l[n]})$  (Type: Phase shift matrix for layer $l$ at time $n$; Domain: $\mathbb{C}^{K \times K}$; Unit: None)
$\mathbf{W}^l$  (Type: Transmission matrix between layers $l-1$ and $l$; Domain: $\mathbb{C}^{K \times K}$; Unit: None)
$\boldsymbol{w}_m^1$  (Type: Transmission vector from ATCo antenna $m$ to first metasurface layer; Domain: $\mathbb{C}^{K \times 1}$; Unit: None)
$\lambda$  (Type: Carrier wavelength; Domain: $\mathbb{R}^+$; Unit: m)
$d_x, d_y$  (Type: Size of a meta-atom along x and y axes; Domain: $\mathbb{R}^+$; Unit: m)
$\boldsymbol{h}_m^H[n]$  (Type: Channel vector from last SIM layer to eVTOL $m$ at time $n$; Domain: $\mathbb{C}^{1 \times K}$; Unit: None)
$\rho_0$  (Type: Reference path loss at 1m; Domain: $\mathbb{R}^+$; Unit: None (often in dB))
$\alpha^h$  (Type: Path loss exponent; Domain: $\mathbb{R}_{\geq 2}$; Unit: None)
$\kappa^h$  (Type: Rician factor; Domain: $\mathbb{R}^+$; Unit: dB) $\sigma_m^2$  (Type: Receiver noise power at eVTOL $m$; Domain: $\mathbb{R}^+$; Unit: W)
$s_m[n]$  (Type: Transmission data symbol for eVTOL $m$ at time $n$; Domain: $\mathbb{C}$; Unit: None; Assumption: $\mathbb{E}\{s_m[n]\} = 0$, $\mathbb{E}\{|s_m[n]|^2\} = 1$, i.i.d.)

**Core Equations**  **1) SIM Beamforming Matrix.**
The end-to-end beamforming matrix $\mathbf{G}[n]$ of the $L$-layer SIM is given by the product of the transmission and phase shift matrices across all layers.

$$\mathbf{G}[n] = \boldsymbol{\Psi}^L[n]\mathbf{W}^L\boldsymbol{\Psi}^{L-1}[n] \cdots \boldsymbol{\Psi}^2[n]\mathbf{W}^2\boldsymbol{\Psi}^1[n] \in \mathbb{C}^{K \times K}$$

---

**2) Inter-layer Transmission Matrix Entry.**

The $(k, k')$-th entry of the transmission matrix $\mathbf{W}^l$ is derived from Rayleigh-Sommerfeld diffraction theory.

$$w_{k,k'}^l = \frac{d_x d_y \cos \chi_{k,k'}^l}{d_{k,k'}^l} \left( \frac{1}{2\pi d_{k,k'}^l} - j\frac{1}{\lambda} \right) e^{j2\pi d_{k,k'}^l / \lambda}$$

where $d_{k,k'}^l$ is the distance between meta-atoms, and $\chi_{k,k'}^l$ is the angle between the propagation direction and the normal to the layer.

**3) Air-Ground Channel Model.**

The channel from the SIM to eVTOL $m$ is modeled as a Rician fading channel. The $k$-th entry is:

$$h_{m,k}[n] = \sqrt{\frac{\rho_0}{(d_m[n])^{\alpha^h}}} \sqrt{\frac{\kappa^h}{\kappa^h + 1}} \bar{h}_m[n]$$

where $d_m[n] = \|\boldsymbol{q}_m[n] - \mathbf{B}\|$ is the distance from the ATCo station to the eVTOL, and $\bar{h}_m[n] = 1$ is assumed for the LoS component.

**4) Received Signal.**

The composite signal received by eVTOL $m$ at time slot $n$ is:

$$y_m[n] = \boldsymbol{h}_m^H[n]\mathbf{G}[n] \sum_{m'=1}^{M} \boldsymbol{w}_{m'}^1 \sqrt{p_{m'}[n]} s_{m'}[n] + \tau$$

where $\tau \sim \mathcal{CN}(0, \sigma_m^2)$ is the complex AWGN.

**5) Signal-to-Interference-plus-Noise Ratio (SINR).**

The SINR for eVTOL $m$ at time $n$ is:

$$\mathrm{SINR}_m[n] = \frac{|\boldsymbol{h}_m^H[n]\mathbf{G}[n]\boldsymbol{w}_m^1|^2 p_m[n]}{\sum_{m'=1, m' \neq m}^{M} |\boldsymbol{h}_{m'}^H[n]\mathbf{G}[n]\boldsymbol{w}_{m'}^1|^2 p_{m'}[n] + \sigma_m^2}$$

**6) Achievable Data Rate.**

The achievable data rate for eVTOL $m$ at time $n$ is given by the Shannon capacity formula:

$$R_m[n] = \log\left(1 + \mathrm{SINR}_m[n]\right)$$

**7) Joint Optimization Problem (P1).**

The overall problem is formulated to maximize the sum rate over all eVTOLs and time slots by jointly optimizing power allocation $\mathbf{P}$, phase shifts $\boldsymbol{\Psi}$, and trajectories $\mathcal{Q}$.

$$(\text{P1}): \quad \max_{\mathbf{P}, \boldsymbol{\Psi}, \mathcal{Q}} g(\mathbf{P}, \boldsymbol{\Psi}, \mathcal{Q}) = \sum_{n=1}^{N} \sum_{m=1}^{M} R_m[n]$$

$$\text{s.t.}$$

$$\mathbf{C1}: \sum_{m=1}^{M} p_m[n] \leq P_{ATC}, \forall n \in N$$

$$\mathbf{C2}: p_m[n] \geq 0, \forall n \in N, \forall m \in M$$

$$\mathbf{C3}: \theta_k^l[n] \in [0, 2\pi), \forall n, k, l$$

$$\mathbf{C4}: \|\boldsymbol{q}_m[n] - \boldsymbol{q}_m[n-1]\| \leq V_{max}\delta, \forall n, m$$

$$\mathbf{C5}: \boldsymbol{q}_m[n] \in \mathcal{R}_{cor}, \forall n, m$$

$$\mathbf{C6}: \boldsymbol{q}_m[0] = \boldsymbol{f}_m[0], \boldsymbol{q}_m[N] = \boldsymbol{f}_m[N], \forall m$$

**8) Composite Potential Field (CPF) Force.**

The flight control acceleration for eVTOL $i$ is derived from the negative gradient of the combined potential fields.

$$\boldsymbol{a}_i[n] = -\nabla \left( \mathcal{F}_i^{com}[n] + \mathcal{F}_i^{sep}[n] + \mathcal{F}_i^{tar}[n] \right)$$

The individual fields (target $\mathcal{F}^{tar}$, separation $\mathcal{F}^{sep}$, communication $\mathcal{F}^{com}$) are functions of the eVTOL's state and hyperparameters $\{k_{tar}, k_{sep}, k_{com}\}$ which are optimized via a DQN framework.

## G.2 EXAMPLE 2: MULTI-UAV PATROL INSPECTION WITH MOBILE EDGE COMPUTING

This system model captures the complexity of joint communication, computation, and trajectory optimization in UAV-enabled MEC networks.

---

### UAV-MEC System Model

**Background**   This paper considers a multi-UAV patrol inspection system where UAVs traverse pre-determined cruise points to collect data and offload it to Ground Base Stations (GBSs) equipped with Mobile Edge Computing (MEC) servers for processing. The system model jointly optimizes cruise point assignment, communication scheduling, computational allocation, and UAV trajectory to minimize total energy consumption and balance task completion times among UAVs.

**Key Assumptions**   • UAVs fly at a constant altitude $H_U$.
• GBSs are deployed with sufficient density to ensure continuous cellular coverage.
• A TDMA scheme is used for UAV-GBS communication.
• The communication rate model incorporates a logistic function of the elevation angle, based on empirical measurements.
• The information causality constraint must be satisfied (processed data $\leq$ received data).
• UAV dynamics follow a rotary-wing energy consumption model.
• The CPU cycles required per bit ($C_U$) are known and depend on the task type.

**Parameter Definitions**

$$\mathcal{U} = \{u_1, ..., u_N\} \quad \text{Set of } N \text{ UAVs}$$
$$\mathcal{G} = \{g_1, ..., g_M\} \quad \text{Set of } M \text{ GBSs}$$
$$\boldsymbol{S} = \{s_1, ..., s_K\} \quad \text{Set of } K \text{ cruise points}$$
$$\mathbf{w}_{s_k} \in \mathbb{R}^{2 \times 1} \quad \text{Coordinates of cruise point } s_k \text{ (m)}$$
$$\mathbf{w}_{g_m} \in \mathbb{R}^{2 \times 1} \quad \text{Coordinates of GBS } g_m \text{ (m)}$$
$$H_U, H_G \quad \text{Altitude of UAV and GBS, respectively (m)}$$
$$\boldsymbol{\eta}(t) \in \mathbb{R}^{2 \times 1} \quad \text{UAV's horizontal position at time } t \text{ (m)}$$
$$\mathbf{v}(t) \quad \text{UAV's velocity vector at time } t \text{ (m/s); } \|\mathbf{v}(t)\| \leq V_{max}$$
$$Q_{s_k} \quad \text{Data volume collected at cruise point } s_k \text{ (bits)}$$
$$R_{g_m}(t) \quad \text{Real-time communication rate to GBS } g_m \text{ (bps)}$$
$$\tau_{g_m}(t) \in \{0, 1\} \quad \text{Binary scheduling indicator for GBS } g_m$$
$$f_U(t), f_{g_m}(t) \quad \text{CPU frequency of UAV and GBS } g_m, \text{ respectively (cycles/s)}$$
$$C_U \quad \text{CPU cycles required per bit (cycles/bit)}$$
$$P(t) \quad \text{UAV transmission power (W)}$$
$$T_i \quad \text{Task completion time for } i\text{-th UAV (s)}$$
$$\vartheta_U \quad \text{UAV's effective capacitance coefficient (F)}$$

**Core Equations**   **1) Distance and Elevation Angle.**
The distance between the UAV and a GBS $g_m$ at time $t$ is:

$$d_{g_m}(t) = \sqrt{(H_U - H_G)^2 + \|\boldsymbol{\eta}(t) - \mathbf{w}_{g_m}\|^2}$$

The corresponding elevation angle is:

$$\theta_{g_m}(t) \triangleq \frac{180}{\pi} \arctan\left(\frac{H_U - H_G}{\|\boldsymbol{\eta}(t) - \mathbf{w}_{g_m}\|}\right)$$

*Assumptions:* LOS propagation is dominant. UAV and GBS altitudes are constant.

*Domain:* $\theta_{g_m}(t) \in (0°, 90°]$, $d_{g_m}(t) > 0$.

**2) Communication Rate Model.**
The real-time communication rate is given by:

$$R_{g_m}(t) = \left(\chi_3 + \frac{\chi_4}{1 + e^{-(\chi_1 + \chi_2 \theta_{g_m}(t))}}\right) H \log_2\left(1 + \frac{\hat{\gamma} P(t)}{(d_{g_m}(t))^\alpha}\right)$$

*Variables/Constants:* $\chi_1, \chi_2, \chi_3, \chi_4$ are environment-dependent parameters ($\chi_1 < 0$, $\chi_2 > 0$, $\chi_4 > 0$, $\chi_3 + \chi_4 = 1$). $H$ is the bandwidth (Hz). $\hat{\gamma} = \beta_0/(\sigma^2 \Lambda)$ is the normalized SNR, where $\beta_0$ is the reference channel gain (dB), $\sigma^2$ is the noise power (W), and $\Lambda$ is the SNR gap. $\alpha$ is the path-loss exponent.

---

*Assumptions:* The model accounts for the practical dependence of antenna gain on the elevation angle.

*Domain:* $R_{g_m}(t) \geq 0$.

**3) Information Causal Constraint.**

The data processed by a GBS cannot exceed the data received from the UAV:

$$\int_0^{T_P} \frac{f_{g_m}(t)}{C_U} dt \leq \int_0^{T_P} \tau_{g_m}(t) R_{g_m}(t) dt, \quad \forall T_P \in [0, T_i]$$

*Assumptions:* No data buffering at the GBS beyond what is received.

*Domain:* $T_P \geq 0$.

**4) Energy Consumption Models.**

The total energy for the $i$-th UAV, $E_i$, is the sum of computation energy ($E_c$), transmission energy ($E_t$), and flight energy ($E_f$).

$$E_t = \sum_{m=1}^{M} \sum_{k=1}^{K_i} \int_0^{T_{s_k}} \tau_{g_m}(t) P(t) dt$$

$$E_c = \int_0^{T_i} \vartheta_U f_U^3(t) dt$$

$$E_f = \int_0^{T_i} \left[ P_0 \left( 1 + \frac{3\|\mathbf{v}(t)\|^2}{U_{tip}^2} \right) + P_i \left( \sqrt{1 + \frac{\|\mathbf{v}(t)\|^4}{4v_0^4} - \frac{\|\mathbf{v}(t)\|^2}{2v_0^2}} \right)^{1/2} + \frac{1}{2} d_0 \rho s \hat{a} \|\mathbf{v}(t)\|^3 \right] dt$$

$$E_i = E_c + E_t + E_f$$

*Variables/Constants:* $P_0, P_i$ are blade profile and induced power in hover (W). $U_{tip}$ is rotor tip speed (m/s). $v_0$ is mean rotor induced velocity in hover (m/s). $d_0$ is fuselage drag ratio. $\rho$ is air density (kg/m³). $s$ is rotor solidity. $\hat{a}$ is rotor disc area (m²).

*Assumptions:* Rotary-wing UAV dynamics. DVFS is used for computation.

*Domain:* $E_t, E_c, E_f, E_i \geq 0$.

**5) Original Optimization Problem (P0).**

The joint optimization problem is formulated as:

$$(\text{P0}): \min_{\substack{\{\pi(k)\}, \{\boldsymbol{\eta}(t)\}, \\ \{\tau_{g_m}(t)\}, t_{s_{\pi(k)}}, T_i, K_i}} \sum_{i=1}^{N} \left( E_i + \phi T_i + \lambda(T_i - T_{avg}) \right)$$

$$\text{s.t. } \tau_{g_m}(t) \in \{0, 1\}, \sum_{m=1}^{M} \tau_{g_m}(t) \leq 1 \quad \forall t \quad (9a)$$

Information causal constraint (4)

Data processing demand: UAV + GBSs must process all collected data $Q_{s_{\pi(k)}}$

Trajectory constraints: Start at $\mathbf{s}_I$, visit all points in $\pi$, end at $\mathbf{s}_F$

Velocity constraint: $\|\mathbf{v}(t)\| \leq V_{max}$

- *Variables/Constants:* $\phi, \lambda$ are compensation factors to balance the dimensions of energy and time in the objective.

- *Assumptions:* The problem is decomposed into two tractable subproblems: Task Assignment and Path Planning.

- *Domain:* The problem is non-convex and requires decomposition for solution.

## G.3 EXAMPLE 3: RIS-AIDED UNSOURCED RANDOM ACCESS

### RIS-Aided URA System

**Background**  The paper proposes a RIS-aided unsourced random access (URA) system where a massive number of users communicate with a base station (BS) via a reconfigurable intelligent surface (RIS). The direct user-BS links are assumed completely blocked, making the RIS essential for connectivity. The system employs a slotted transmission structure with joint pilot detection, channel estimation, and RIS phase shift optimization to enable reliable communication.

**Key Assumptions**
- Quasi-static block fading channels (constant over a frame)
- Perfect knowledge of RIS-BS channel $\mathbf{G}$ (stationary elements)
- Passive RIS with unit-modulus phase shifts: $|[\mathbf{w}_t]_i| = 1$
- Blocked direct user-BS links (no direct path)
- Saleh-Valenzuela channel model for RIS-BS and user-RIS links
- UPA antenna arrays at both BS and RIS

**Parameter Definitions**  $\mathbf{G} \in \mathbb{C}^{M \times N}$  (RIS-BS channel matrix; Type: Geometric; Unit: dimensionless)
$\mathbf{h}_i \in \mathbb{C}^{1 \times N}$  (User-RIS channel vector for user $i$; Type: Geometric; Unit: dimensionless)
$\mathbf{w}_t \in \mathbb{C}^{N \times 1}$  (RIS phase shift vector at time $t$; Type: Control; Domain: $|[\mathbf{w}_t]_i| = 1$)
$x_{i,t} \in \mathbb{C}$  (Transmitted symbol from user $i$ at time $t$; Type: Information; Unit: dimensionless)
$\mathbf{z}_t \in \mathbb{C}^{M \times 1}$  (Noise vector; Type: AWGN; Distribution: $\mathcal{CN}(\mathbf{0}, \sigma_z^2 \mathbf{I}_M)$)
$M$  (Number of BS antennas; Type: Integer; Unit: dimensionless)
$N$  (Number of RIS elements; Type: Integer; Unit: dimensionless)
$K_a$  (Number of active users; Type: Integer; Unit: dimensionless)
$n$  (Total channel uses; Type: Integer; Unit: dimensionless)
$L_G$  (Number of paths in RIS-BS channel; Type: Integer; Unit: dimensionless)
$L_{R,i}$  (Number of paths in user-RIS channel; Type: Integer; Unit: dimensionless)

**Core Equations**

**1) Received Signal Model (Eq. 4):**

$$\mathbf{y}_t = \sum_{i=1}^{K_a} \mathbf{G}\text{diag}(\mathbf{h}_i)\mathbf{w}_t x_{i,t} + \mathbf{z}_t, \quad t = 1, \ldots, n$$

*Variables:* $\mathbf{y}_t \in \mathbb{C}^{M \times 1}$ (received signal),
$\quad \mathbf{G} \in \mathbb{C}^{M \times N}, \mathbf{h}_i \in \mathbb{C}^{1 \times N}, \mathbf{w}_t \in \mathbb{C}^{N \times 1}, x_{i,t} \in \mathbb{C}, \mathbf{z}_t \in \mathbb{C}^{M \times 1}$
*Assumptions: Blocked direct links, passive RIS, quasi-static channels*
*Domain:* $|[\mathbf{w}_t]_i| = 1, \quad t \in \{1, \ldots, n\}$

**2) Pilot Phase Received Signal (Eq. 5):**

$$\mathbf{Y}_p = \sqrt{P_p} \sum_{i \in \mathcal{S}_s} \mathbf{G}\text{diag}(\mathbf{h}_i)\mathbf{W}_{p_s}\text{diag}(\mathbf{p}_i) + \mathbf{Z}_p$$

*Variables:* $\mathbf{Y}_p \in \mathbb{C}^{M \times n_p}, \mathbf{W}_{p_s} \in \mathbb{C}^{N \times n_p}, \mathbf{p}_i \in \mathbb{C}^{1 \times n_p}, \mathbf{Z}_p \in \mathbb{C}^{M \times n_p}$
*Assumptions: Fixed RIS configuration during pilot phase*
*Domain:* $|[\mathbf{W}_{p_s}]_{i,j}| = 1$

**3) Data Phase Received Signal (Eq. 6):**

$$\mathbf{Y}_{c,f} = \sqrt{P_c} \sum_{i \in \mathcal{S}_s} \mathbf{G}\text{diag}(\mathbf{h}_i)\mathbf{W}_{c_s}\text{diag}(\mathbf{b}_i)v_{i,f} + \mathbf{Z}_{c,f}$$

*Variables:* $\mathbf{Y}_{c,f} \in \mathbb{C}^{M \times n_s}, \mathbf{W}_{c_s} \in \mathbb{C}^{N \times n_s}, \mathbf{b}_i \in \mathbb{C}^{1 \times n_s}, v_{i,f} \in \{\pm 1\}$
*Assumptions: Two RIS configurations $\mathcal{C}_0$ (constant) and $\mathcal{C}_1$ (varying)*
*Domain:* $|[\mathbf{W}_{c_s}]_{i,j}| = 1, \quad v_{i,f} \in \{\pm 1\}$

**4) Channel Model - RIS-BS (Eq. 1):**

$$\mathbf{G} = \sqrt{MN} \sum_{l=1}^{L_G} \mu_l \mathbf{a}_M(\phi_{r,l}, \psi_{r,l})^T \mathbf{a}_N(\phi_{t,l}, \psi_{t,l})$$

*Variables:* $\mu_l \sim \mathcal{CN}(0, L_0 d_l^{-\alpha_{PL}})$, $\mathbf{a}_M(\cdot), \mathbf{a}_N(\cdot)$ *(steering vectors)*

*Assumptions: Saleh-Valenzuela model, UPA arrays*

*Domain:* $\phi_{r,l}, \psi_{r,l} \in [0, 2\pi), \phi_{t,l}, \psi_{t,l} \in [0, 2\pi)$

**5) Channel Model - User-RIS (Eq. 3):**

$$\mathbf{h}_i = \sqrt{N} \sum_{f_i=1}^{L_{R,i}} \mu_{f_i} \mathbf{a}_N(\phi_{i,f_i}, \psi_{i,f_i})$$

*Variables:* $\mu_{f_i} \sim \mathcal{CN}(0, L_0 d_{f_i}^{-\alpha_{PL}})$

*Assumptions: Same path loss model as RIS-BS channel*

*Domain:* $\phi_{i,f_i}, \psi_{i,f_i} \in [0, 2\pi)$

**6) Steering Vector Model (Eq. 2):**

$$\mathbf{a}_N(\phi, \psi) = \frac{1}{\sqrt{N}} e^{-j2\pi\bar{\phi}\mathbf{n_1}} \otimes e^{-j2\pi\bar{\psi}\mathbf{n_2}}$$

*Variables:* $\bar{\phi} = \sin(\phi)\cos(\psi), \bar{\psi} = \sin(\psi)$

$$\mathbf{n}_1 = \frac{d}{\lambda}[0, \dots, N_1 - 1], \mathbf{n}_2 = \frac{d}{\lambda}[0, \dots, N_2 - 1]$$

*Assumptions: UPA structure with antenna spacing $d = \lambda/2$*

*Domain:* $\phi, \psi \in [0, 2\pi)$

**Model Validation**

- *Dimensional consistency:* All matrix multiplications are dimensionally consistent (e.g., $\mathbf{G} \in \mathbb{C}^{M \times N}$ multiplied by $\mathrm{diag}(\mathbf{h}_i) \in \mathbb{C}^{N \times N}$ yields $\mathbb{C}^{M \times N}$ matrix)
- *Boundary conditions:* Unit-modulus constraint $|[\mathbf{w}_t]_i| = 1$ enforced for passive RIS
- *Parameter homogeneity:* All channel gains $\mu_l, \mu_{f_i}$ have consistent units (dimensionless with path loss scaling)
- *Physical constraints:* Angle parameters restricted to $[0, 2\pi)$, array steering vectors properly normalized

## G.4 MODEL EXTRACTION QUALITY ASSESSMENT

These extracted models demonstrate several quality indicators that validate our automated extraction pipeline:

**Completeness:** Each model includes comprehensive variable definitions with proper units and domains, ensuring self-contained mathematical descriptions suitable for question generation.

**Mathematical Rigor:** The extractions preserve complex mathematical relationships including multi-layer matrix products, integral constraints, and summation indices, maintaining the precision required for technical education.

**Domain Coverage:** The three examples span classical communication theory (Shannon capacity), modern optimization frameworks (joint resource allocation), and emerging technologies (RIS, SIM), reflecting the breadth of WirelessMathBench-XL.

**Hierarchical Structure:** Models successfully capture equation dependencies, from basic distance calculations to complex optimization objectives, enabling progressive question difficulty design.

# H HUMAN EXPERT EVALUATION EXAMPLES

This section presents representative examples from our expert evaluation process, demonstrating the application of our quality rubric across different score levels. Each example includes the complete question as presented to evaluators, with expert annotations highlighting strengths and weaknesses.

## H.1 SCORE 5 - EXCELLENT QUALITY

Questions scoring 5 demonstrate comprehensive variable definitions, clear mathematical structure, and strong pedagogical value. These questions are ready for immediate use in educational or evaluation contexts.

---

### Question ID: 14024

**Paper:** 2508.03740v1

**Answer:** $\mathcal{L}_{VQ} = \|\text{sg}[\mathbf{F}] - \mathbf{C}\|_2^2 + \alpha \|\mathbf{F} - \text{sg}[\mathbf{C}]\|_2^2 + \beta D_{KL}(p_c\|p_u)$

BACKGROUND

In the vector quantization training process, a composite loss function ensures proper codebook learning and feature quantization. The loss consists of three components: codebook loss, commitment loss, and usage regularization, where $\mathbf{F} \in \mathbb{R}^{M \times K}$ is the semantic feature matrix, $\mathbf{C} \in \mathbb{R}^{N \times K}$ is the codebook matrix, $\text{sg}[\cdot]$ denotes the stop-gradient operator, $D_{KL}(\cdot)$ is the Kullback-Leibler divergence, $p_c$ is the codeword usage distribution, $p_u$ is the uniform distribution, and $\alpha, \beta \in \mathbb{R}^+$ are hyperparameters that weight the different loss components.

QUESTION

*Write the complete vector quantization loss function with all three components.*

EQUATION

$$[MASK]$$

— Please solve this problem step by step. Fill in the [MASK] placeholder(s) with the correct mathematical expression(s). Your final answer should be given at the end in the format: $your\_answer$

---

**Question ID: 14134**

**Paper:** 2206.08306v1

**Answer:**
$$\frac{\left(m\frac{dv}{dt} + \frac{1}{2}\rho_{air}A_fC_Dv^2 + mgr_0\cos(\alpha) + mg\sin(\alpha)\right)v/\eta_t + P_{accessories}}{\eta_e}$$

BACKGROUND

The instantaneous fuel rate model calculates the mass flow rate of fuel consumed. Here, $\dot{m}_f$ is the fuel rate (in kg/s), $m$ is the vehicle mass (in kg), $\frac{dv}{dt}$ is the acceleration (in m/s²), $\rho_{air}$ is the air density (in kg/m³), $A_f$ is the frontal area (in m²), $C_D$ is the drag coefficient (dimensionless), $v$ is the speed (in m/s), $g$ is gravitational acceleration (in m/s²), $r_0$ is the rolling resistance coefficient (dimensionless), $\alpha$ is the road grade (in radians), $\eta_t$ is the transmission efficiency (dimensionless), $P_{accessories}$ is the power consumed by vehicle accessories (in W), and $\eta_e$ is the engine efficiency (dimensionless).

QUESTION

*Write the complete equation for the instantaneous fuel rate.*

EQUATION

$$\dot{m}_f = [MASK]$$

— Please solve this problem step by step. Fill in the [MASK] placeholder(s) with the correct mathematical expression(s). Your final answer should be given at the end in the format: $\boxed{your\_answer}$

## Question ID: 4149

**Paper:** 2505.19983v1

**Answer:** 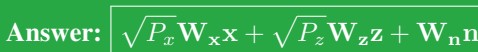 $\sqrt{P_x}\mathbf{W_x}\mathbf{x} + \sqrt{P_z}\mathbf{W_z}\mathbf{z} + \mathbf{W_n}\mathbf{n}$

BACKGROUND

In a wireless semantic communication system with interference, the received real-valued signal after equalization combines the desired signal, an interference signal, and noise. The system model is derived from the complex baseband representation, where $\mathbf{y} \in \mathbb{R}^{2k}$ is the equalized received real signal vector, $\mathbf{x} \in \mathbb{R}^{2k}$ is the real-valued semantic feature vector to be transmitted, $\mathbf{z} \in \mathbb{R}^{2k}$ is the real-valued interference vector, $\mathbf{n} \sim \mathcal{N}(0, \frac{\sigma^2}{2}\mathbf{I}_{2k})$ is the real-valued additive white Gaussian noise vector, $P_x \in \mathbb{R}^+$ is the desired signal transmit power (in linear scale), $P_z \in \mathbb{R}^+$ is the interference signal transmit power (in linear scale), $\mathbf{W_x} \in \mathbb{R}^{2k \times 2k}$ is the channel transformation matrix for the desired signal, $\mathbf{W_z} \in \mathbb{R}^{2k \times 2k}$ is the channel transformation matrix for the interference signal, and $\mathbf{W_n} \in \mathbb{R}^{2k \times 2k}$ is the channel transformation matrix for the noise.

QUESTION

*Write the complete received signal equation including all three components.*

EQUATION

$$\mathbf{y} = [MASK]$$

— Please solve this problem step by step. Fill in the [MASK] placeholder(s) with the correct mathematical expression(s). Your final answer should be given at the end in the format: $\boxed{your\_answer}$

## H.2 SCORE 4 - GOOD QUALITY

Questions scoring 4 contain solid technical content with minor areas for improvement, typically in completeness of context or clarity of problem statement.

---

**Question ID: 14101**

**Paper:** 2208.11967v1

**Answer:** 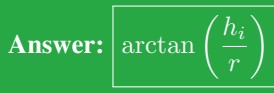 $\arctan\left(\dfrac{h_i}{r}\right)$

BACKGROUND

In a laser-powered UAV-assisted wireless network, the probability of having a Line-of-Sight (LOS) link is crucial for signal propagation. This model characterizes the LOS probability between an aerial or terrestrial node and a user, where $\mathfrak{P}_i(r)$ is the probability of an LOS link for node type $i$ (where $i \in \{\text{Lu}, \text{Lb}\}$ representing LOS UAV and LOS TBS links, respectively), $r$ is the horizontal distance between the transmitter and receiver (in meters), $h_i$ is the altitude or height of the node type $i$ (in meters), and $a$, $b$, $c$ are environment-dependent parameters (dimensionless) that model the blockage characteristics in urban, suburban, or dense urban environments.

QUESTION

*What trigonometric function of the elevation angle is the argument of the exponential?*

EQUATION

$$\mathfrak{P}_i(r) = -a \exp\left(-b[MASK]\right) + c$$

— Please solve this problem step by step. Fill in the [MASK] placeholder(s) with the correct mathematical expression(s). Your final answer should be given at the end in the format: $\boxed{your\_answer}$

---

## Question ID: 4439

**Paper:** 2506.01400v1

**Answer:** 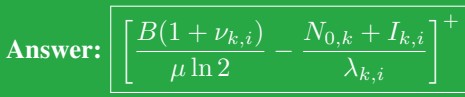

BACKGROUND

The optimal power allocation for communication UEs in a multi-user MIMO system is derived using the Karush-Kuhn-Tucker (KKT) conditions to solve the constrained optimization problem. This solution follows a water-filling structure. Here, $P_{C,k,i}$ is the optimal power allocated to the $i$-th sub-channel of communication UE $k$ (in W), $[\cdot]^+ = \max(0, \cdot)$ ensures non-negative power, $B$ is the bandwidth (in Hz), $\nu_{k,i}$ is the Lagrange multiplier associated with the minimum capacity constraint for the $i$-th sub-channel of UE $k$ (dimensionless), $\mu$ is the Lagrange multiplier associated with the total power constraint (in W$^{-1}$), $N_{0,k}$ is the noise power at UE $k$ (in W), $I_{k,i}$ is the interference power (in W), and $\lambda_{k,i}$ is the channel gain eigenvalue (dimensionless).

QUESTION

*Write the complete optimal power allocation formula for a communication user equipment (UE).*

EQUATION

$$P_{C,k,i} = [MASK]$$

— Please solve this problem step by step. Fill in the [MASK] placeholder(s) with the correct mathematical expression(s). Your final answer should be given at the end in the format: $your\_answer$

## H.3 SCORE 3 - ACCEPTABLE QUALITY

Questions scoring 3 meet minimum requirements but have noticeable gaps in clarity or completeness that limit their educational value.

---

**Question ID: 13890**

**Paper:** 2208.07045v1

Answer: A

BACKGROUND

In an interference-coupled multi-cell RAN slicing system, the Signal-to-Interference-plus-Noise Ratio (SINR) is calculated at a specific user location. The SINR determines the quality of the wireless link for a user served by a particular channel in a slice, where $\gamma_{s,q}(l, \mathbf{\Delta}_{s,q})$ is the SINR at location $l$ for channel $q$ in slice $s$ (dimensionless), $P_{s,q}^{\mathrm{SI}}(l)$ is the received signal power at location $l$ from the base station transmitting on channel $q$ in slice $s$ (in watts), $\mathcal{N}_{s,q}$ is the set of all slice-channel pairs that can potentially interfere with $(s,q)$ (dimensionless), $\mathbf{\Delta}_{s,q}$ is a binary vector indicating which interfering transmitters in $\mathcal{N}_{s,q}$ are active (dimensionless), $P_{(s',q'),(s,q)}^{\mathrm{IN}}(l)$ is the interference power at location $l$ from an interfering transmitter on channel $q'$ in slice $s'$ (in watts), and $N_0$ is the noise power (in watts).

QUESTION

*Which expression correctly represents the SINR calculation that should replace [MASK]?*

EQUATION

$$\gamma_{s,q}(l, \mathbf{\Delta}_{s,q}) = [MASK]$$

OPTIONS

- **A:** $\dfrac{P_{s,q}^{\mathrm{SI}}(l)}{\displaystyle\sum_{(s',q')\in\mathcal{N}_{s,q}\setminus(s,q)} \mathbf{\Delta}_{s,q}(s',q')P_{(s',q'),(s,q)}^{\mathrm{IN}}(l) + N_0}$

- B: $\dfrac{P_{s,q}^{\mathrm{SI}}(l)}{\displaystyle\sum_{(s',q')\in\mathcal{N}_{s,q}} \mathbf{\Delta}_{s,q}(s',q')P_{(s',q'),(s,q)}^{\mathrm{IN}}(l) + N_0}$

- C: $\dfrac{P_{s,q}^{\mathrm{SI}}(l)}{\displaystyle\sum_{(s',q')\in\mathcal{N}_{s,q}\setminus(s,q)} P_{(s',q'),(s,q)}^{\mathrm{IN}}(l) + N_0}$

- D: $\dfrac{\displaystyle\sum_{(s',q')\in\mathcal{N}_{s,q}\setminus(s,q)} \mathbf{\Delta}_{s,q}(s',q')P_{(s',q'),(s,q)}^{\mathrm{IN}}(l)}{P_{s,q}^{\mathrm{SI}}(l) + N_0}$

- Your final answer should be given at the end in the format: $\boxed{X}$

---

## Question ID: 4275

**Paper:** 2504.18155v1      **Answer: A**

BACKGROUND

In the hierarchical cell-free massive MIMO uplink training phase, edge access points (eAPs) receive pilot sequences from multiple users. The received pilot signal matrix at eAP $l$ combines contributions from all users through their respective channels, where $\mathbf{\Psi}_l \in \mathbb{C}^{N_a \times \tau_p}$ represents the received pilot signal matrix at eAP $l$, $p_u$ is the user transmit power constraint (in watts), $\mathbb{K} = \{1, \dots, K\}$ is the set of user indices, $\mathbf{h}_{kl} \in \mathbb{C}^{N_a \times 1}$ is the channel vector from user $k$ to eAP $l$, $\mathbf{i}_k \in \mathbb{C}^{\tau_p \times 1}$ is the pilot sequence of user $k$ (dimensionless), $\mathbf{Z}_l \in \mathbb{C}^{N_a \times \tau_p}$ is the additive noise matrix with entries $\sim \mathcal{CN}(0, \sigma_z^2)$, $N_a$ is the number of antennas per eAP, and $\tau_p$ is the pilot sequence length (in symbols).

QUESTION

*Which expression correctly represents the received pilot signal matrix at eAP $l$?*

EQUATION

$$\mathbf{\Psi}_l = [MASK]$$

OPTIONS

- **A: $\sqrt{p_u} \sum_{k \in \mathbb{K}} \mathbf{h}_{kl} \mathbf{i}_k^T + \mathbf{Z}_l$**

- B: $\sqrt{p_u} \sum_{k \in \mathbb{K}} \mathbf{h}_{kl} \mathbf{i}_k^H + \mathbf{Z}_l$

- C: $\sqrt{p_u} \sum_{k \in \mathbb{K}} \mathbf{h}_{kl}^T \mathbf{i}_k + \mathbf{Z}_l$

- D: $\sqrt{p_u} \sum_{k \in \mathbb{K}} \mathbf{h}_{kl}^H \mathbf{i}_k + \mathbf{Z}_l$

- Your final answer should be given at the end in the format: $\boxed{X}$

## H.4 SCORE 2 - POOR QUALITY

Questions scoring 2 have significant deficiencies that impair their usefulness, though they may contain salvageable elements.

---

**Question ID: 13936**

**Paper:** 2502.11053v2

**Answer:** $\phi$

BACKGROUND

In the belief propagation decoding of LDPC codes, messages are passed between nodes on the Tanner graph. For the check node update, $\mathcal{L}(\mathbf{r}_{ji}) \in \mathbb{R}$ is the log-likelihood ratio (LLR) message sent from check node $j$ to bit node $i$. $\mathcal{L}(\mathbf{q}_{i'j}) \in \mathbb{R}$ is the LLR message received from a connected bit node $i'$. The set $\mathrm{BN}_{j \setminus i}$ contains all bit nodes connected to check node $j$ except bit node $i$.

QUESTION

*Which function is applied to the absolute value of each incoming LLR before summation in the stable SPA update?*

EQUATION

$$\mathcal{L}(\mathbf{r}_{ji}) = \left( \prod_{i' \in \mathrm{BN}_{j \setminus i}} \mathrm{sign}\left( \mathcal{L}(\mathbf{q}_{i'j}) \right) \right) \cdot \phi \left( \sum_{i' \in \mathrm{BN}_{j \setminus i}} [MASK] \right)$$

— Please solve this problem step by step. Fill in the [MASK] placeholder(s) with the correct mathematical expression(s). Your final answer should be given at the end in the format: $\boxed{your\_answer}$

---

## Question ID: 4173

**Paper:** 2505.18534v1      **Answer:** $\boxed{I(t)}$, $\boxed{\cos}$, $\boxed{\sin}$

BACKGROUND

In a DSP-free coherent optical interconnect system using offset-QAM modulation, the received in-phase and quadrature signals are processed before carrier phase recovery. The system aims to compensate for a phase error between the received signal and the local oscillator. Here, $I'(t)$ represents the received in-phase signal after mixing and before phase correction (in volts or amperes), $Q'(t)$ is the corresponding quadrature signal (in volts or amperes), $I(t) \in \{\pm A_{OMA}/2\}$ is the original modulated in-phase data signal (in volts or amperes), $Q(t) \in \{\pm A_{OMA}/2\}$ is the original modulated quadrature data signal (in volts or amperes), $A_0 \in \mathbb{R}^+$ is the constant DC offset introduced by the offset-QAM modulation format (in volts or amperes), and $\Delta\phi \in (-\pi, \pi]$ is the phase error between the transmitter and local oscillator paths (in radians).

QUESTION

*Complete the three missing components: the data signal and the two trigonometric functions.*

EQUATION

$$I'(t) = ([MASK] + A_0)[MASK](\Delta\phi) + (Q(t) + A_0)[MASK](\Delta\phi)$$

— Please solve this problem step by step. Fill in the [MASK] placeholder(s) with the correct mathematical expression(s). Your final answers should be given at the end in the format: $\boxed{answer1}$, $\boxed{answer2}$, ... (for the 3 blanks in order)

## H.5 Score 1 - Very Poor Quality

Questions scoring 1 have fundamental errors or omissions that render them unusable without complete revision.

---

**Question ID: 13863**

**Paper:** 2412.01187v1

**Answer:** $\boxed{\log}$

BACKGROUND

In a point-to-point, interference-free multi-terminal wireless system with $N_U$ single-antenna users communicating over parallel links, the instantaneous achievable rate is modeled for each link. The rate for terminal $i$ is a function of the channel state and the allocated power, where $r_i(p_i(\boldsymbol{h}), h_i)$ represents the instantaneous achievable rate on link $i$ (in bps/Hz), $p_i(\boldsymbol{h})$ is the power allocated to terminal $i$ for a given channel realization $\boldsymbol{h}$ (in watts), $h_i$ is the fading channel coefficient for terminal $i$ (dimensionless), and $\sigma_i^2$ is the noise variance on link $i$ (in watts). The system assumes AWGN channels and perfect Channel State Information (CSI).

QUESTION

*What is the outer function that transforms the SNR into a rate?*

EQUATION

$$r_i(p_i(\boldsymbol{h}), h_i) \triangleq [MASK]\left(1 + \frac{p_i(\boldsymbol{h}) \cdot h_i^2}{\sigma_i^2}\right)$$

— Please solve this problem step by step. Fill in the [MASK] placeholder(s) with the correct mathematical expression(s). Your final answer should be given at the end in the format: $\boxed{your\_answer}$

---

## Question ID: 4264

**Paper:** 2504.21128v1

**Answer:** 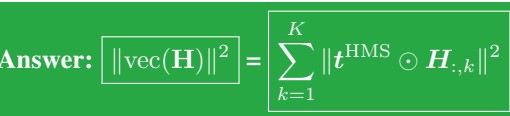

$$\boxed{\|\text{vec}(\mathbf{H})\|^2} = \sum_{k=1}^{K} \|\boldsymbol{t}^{\text{HMS}} \odot \boldsymbol{H}_{:,k}\|^2$$

BACKGROUND

The HMA system must obey a global power conservation constraint for the passive meta-surface. This constraint links the HMS configuration to the communication channel. $\text{vec}(\boldsymbol{H}) \in \mathbb{C}^{NK \times 1}$ is the vectorized channel matrix, $\boldsymbol{t}^{\text{HMS}} \in \mathbb{C}^{N \times 1}$ is the vector of complex transmission coefficients for the $N$ HMS unit-cells, $\boldsymbol{H}_{:,k} \in \mathbb{C}^{N \times 1}$ is the $k$-th column of the channel matrix (representing the channel from user $k$ to all unit-cells), and $\odot$ denotes the Hadamard (element-wise) product.

QUESTION

*Write the complete global power conservation constraint equation.*

EQUATION

$$[MASK] = [MASK]$$

— Please solve this problem step by step. Fill in the [MASK] placeholder(s) with the correct mathematical expression(s). Your final answers should be given at the end in the format: $\boxed{answer1}$, $\boxed{answer2}$, ... (for the 2 blanks in order)

# I  REPRESENTATIVE SOLUTION EXAMPLES FROM WIRELESSMATHLM-7B

This appendix presents detailed examples of solutions generated by WirelessMathLM-7B, organized by question type and quality level. These examples illustrate the model's reasoning patterns, mathematical sophistication, and common error modes identified in our qualitative analysis.

## I.1  HIGH-QUALITY SOLUTION EXAMPLES

### I.1.1  MULTIPLE CHOICE QUESTION: MATRIX ALL-PASS FILTER

**Question ID:** 11325  **Correct Answer:** B  **Model Response:** B

---

**WirelessMathLM-7B Solution**

To determine the correct factorization of the matrix all-pass filter $\mathbf{G}(z)$, we need to understand the properties and definition of a matrix all-pass filter. A matrix all-pass filter is a filter whose frequency response has a magnitude of 1 for all frequencies, but its phase response can vary. Mathematically, a matrix all-pass filter can be represented as:

$$\mathbf{G}(z) = \mathbf{N}(z)\mathbf{D}^{-1}(z)$$

where $\mathbf{N}(z)$ is the numerator matrix polynomial and $\mathbf{D}(z)$ is the denominator matrix polynomial. The matrix all-pass filter is defined such that:

$$\mathbf{G}(z)\mathbf{G}^{-1}(z) = \mathbf{I}_m$$

Given that $\mathbf{D}_0 = \mathbf{I}_m$, the filter is causal and proper. This means that the denominator polynomial $\mathbf{D}(z)$ is a proper polynomial (its degree is less than or equal to $m - 1$).
To verify that the given factorization is correct, we can check the inverse of $\mathbf{G}(z)$:

$$\mathbf{G}^{-1}(z) = (\mathbf{N}(z)\mathbf{D}^{-1}(z))^{-1} = \mathbf{D}(z)\mathbf{N}^{-1}(z)$$

This is because the inverse of a product of matrices is the product of their inverses in reverse order. Now, we can check the product $\mathbf{G}(z)\mathbf{G}^{-1}(z)$:

$$\mathbf{G}(z)\mathbf{G}^{-1}(z) = (\mathbf{N}(z)\mathbf{D}^{-1}(z))(\mathbf{D}(z)\mathbf{N}^{-1}(z))$$
$$= \mathbf{N}(z)\mathbf{D}^{-1}(z)\mathbf{D}(z)\mathbf{N}^{-1}(z) = \mathbf{N}(z)\mathbf{N}^{-1}(z) = \mathbf{I}_m$$

This confirms that $\mathbf{G}(z)$ is indeed an all-pass filter. Therefore, the correct factorization of the matrix all-pass filter $\mathbf{G}(z)$ is:

$$\mathbf{G}(z) = \mathbf{N}(z)\mathbf{D}^{-1}(z)$$

Thus, the correct answer is: $\boxed{B}$

---

**Analysis:** This solution demonstrates sophisticated understanding of matrix theory and filter design. The model correctly identifies the all-pass property, provides mathematical verification through matrix inverse operations, and connects the factorization to causality constraints. The reasoning is systematic, mathematically rigorous, and includes proper justification for each step.

I.1.2   FILL-IN-THE-BLANK (100%): CELL-FREE MASSIVE MIMO BEAMFORMING

**Question ID:** 18369    **Correct Answer:** $\boxed{\sqrt{P_m} \sum_{k=1}^{K} \sqrt{\eta_{mk}} \hat{g}_{mk}^* u_k}$

---

**WirelessMathLM-7B Solution**

To derive the conjugate beamforming transmitted signal in a Cell-Free Massive MIMO (CFmMIMO) system, we need to consider the linear combination of the users' data symbols, precoded using the locally estimated channel state information. The signal transmitted from the $m$-th AP is given by:

$$s_m = \sum_{k \in \mathcal{K}} \eta_{mk} \hat{g}_{mk}^* u_k$$

where: - $s_m \in \mathbb{C}$ is the transmitted symbol from the $m$-th AP (dimensionless), - $P_m \in \mathbb{R}^+$ is the maximum transmit power of the $m$-th AP (in watts, W), - $\eta_{mk} \in \mathbb{R}^+$ is the power control coefficient for user $k$ at AP $m$ (dimensionless), - $\hat{g}_{mk} \in \mathbb{C}$ is the estimated channel coefficient from AP $m$ to user $k$ (dimensionless), - $u_k \in \mathbb{C}$ is the information symbol intended for user $k$ (dimensionless; $\mathbb{E}[|u_k|^2] = 1$).

The term $\hat{g}_{mk}^*$ represents the complex conjugate of the estimated channel coefficient from AP $m$ to user $k$. This is because conjugate beamforming is used to cancel out the phase shifts introduced by the channel.

Therefore, the complete equation for the conjugate beamforming transmitted signal is:

$$s_m = \sum_{k \in \mathcal{K}} \eta_{mk} \hat{g}_{mk}^* u_k$$

So, the final answer is: $\boxed{\sum_{k \in \mathcal{K}} \eta_{mk} \hat{g}_{mk}^* u_k}$

---

**Analysis:** This solution exhibits deep understanding of MIMO systems, correctly identifying conjugate beamforming principles and providing clear physical intuition. The model properly handles complex notation, explains the rationale for complex conjugation, and maintains dimensional consistency throughout the derivation.

### I.1.3 FILL-IN-THE-BLANK (50%): GAUSSIAN FUNCTION COMPONENTS

**Question ID:** 5582    **Correct Answer:** $\boxed{(\lambda - \lambda_p)^2}$ and $\boxed{\Delta\lambda^2}$

---

**WirelessMathLM-7B Solution (Truncated)**

To solve the problem, we need to understand the form of the Gaussian function and how it is modified to include the secondary skewed Gaussian component. The given Gaussian function is:

$$g(\lambda, \lambda_p, \Delta\lambda) = \exp\left[-\frac{[\text{MASK}]}{[\text{MASK}]}\right]$$

The standard form of a Gaussian function centered at $\lambda_p$ with a width parameter $\Delta\lambda$ is:

$$g(\lambda, \lambda_p, \Delta\lambda) = \exp\left[-\frac{(\lambda - \lambda_p)^2}{2(\Delta\lambda)^2}\right]$$

However, the problem involves a secondary skewed Gaussian component, which is typically represented by a function that is asymmetric around the peak wavelength $\lambda_p$...

Given the form of the Gaussian function, the [MASK] placeholder in the exponent should be filled with $(\lambda - \lambda_p)^2$. Therefore, the final answer is:

$\boxed{(\lambda - \lambda_p)^2}$, $\boxed{(\Delta\lambda)^2}$

---

**Analysis:** The model correctly identifies standard Gaussian form and provides appropriate mathematical expressions. The reasoning demonstrates understanding of probability density functions and their parameters in optical communication contexts.

## I.2 ERROR ANALYSIS EXAMPLES

### I.2.1 MATHEMATICAL EQUIVALENCE ERROR

**Question ID:** 2406    **Type:** Mathematical Equivalence Failure

**Correct Answer:** $\boxed{\dfrac{G}{2} - 1}$    **Model Response:** $\boxed{G}$

---

**Error Example: Mathematical Equivalence**

To determine the exponent of the sinusoidal term that defines the pattern's shape, we need to analyze the given equation for the normalized power radiation pattern of a single IRS element:

$$F(\Pi) \triangleq \begin{cases} (\sin\theta\cos\varphi)^{[MASK]}, & \theta \in [0,\pi], \varphi \in [-\frac{\pi}{2}, \frac{\pi}{2}], \\ 0, & \text{otherwise.} \end{cases}$$

The equation suggests that the power radiation pattern is a function of the angles $\theta$ (elevation angle) and $\varphi$ (azimuth angle). The term $\sin\theta\cos\varphi$ is a product of two sinusoidal functions, and the exponent of this product is what we need to identify.

Given that the problem involves a sinusoidal term, we can infer that the exponent is likely a constant that determines the shape of the pattern. Since the problem does not provide any additional information about the specific shape or the value of the exponent, we can assume that the simplest form of the pattern is a first-order sinusoidal function.

Therefore, the exponent of the sinusoidal term is $G$, where $G$ is the peak power gain of a single IRS element. This means that the power radiation pattern is proportional to the square of the sine of the elevation angle and the cosine of the azimuth angle, raised to the power of $G$.

Thus, the exponent of the sinusoidal term is $\boxed{G}$.

---

**Error Analysis:** The model provides reasonable physical interpretation but fails to derive the precise mathematical relationship $\frac{G}{2} - 1$. This represents a common error type where domain knowledge is correctly applied but mathematical transformation is incomplete. The model recognizes that $G$ relates to the exponent but doesn't perform the necessary algebraic manipulation.

### I.2.2 CONCEPTUAL MISUNDERSTANDING ERROR

**Question ID:** 16144     **Type:** Conceptual Misunderstanding

**Correct Answer:**
$$\boxed{\sum_{m=1}^{M} \sum_{k=1}^{K_i} \int_{0}^{T_{s_k}} \tau_{g_m}(t) P(t) dt}$$

---

**Error Example: Conceptual Misunderstanding**

To determine the total transmission energy $E_t$ for a UAV, we need to consider the energy consumed by each transmission to each GBS. The energy consumed by a transmission is given by the product of the transmission power and the time spent transmitting. Therefore, the total transmission energy can be calculated by summing up the energy consumed for all transmissions to all GBSs.

The energy consumed by a transmission from the $i$-th UAV to the $m$-th GBS is $P(t) \cdot T_{s_k} \cdot \tau_{g_m}(t)$, where $P(t)$ is the transmission power, $T_{s_k}$ is the time spent at the $k$-th cruise point, and $\tau_{g_m}(t)$ is the binary scheduling variable...

Thus, the total transmission energy $E_t$ is given by:

$$E_t = \sum_{i=1}^{N} \sum_{k=1}^{K_i} \sum_{m=1}^{M} P(t) \cdot T_{s_k} \cdot \tau_{g_m}(t)$$

The final answer is:
$$\boxed{\sum_{i=1}^{N} \sum_{k=1}^{K_i} \sum_{m=1}^{M} P(t) \cdot T_{s_k} \cdot \tau_{g_m}(t)}$$

---

**Error Analysis:** This error demonstrates correct energy calculation principles but incorrect mathematical formulation. The model understands that energy equals power times time but fails to recognize the need for temporal integration rather than discrete summation. The error reflects misunderstanding of continuous vs. discrete system modeling rather than fundamental energy concepts.

### I.2.3 MCQ SELECTION ERROR

**Question ID:** 16315 **Type:** Multiple Choice Selection

**Correct Answer:** B **Model Response:** C

---

**Error Example: MCQ Selection (Truncated)**

To determine the correct term that completes the equation for the phase compensation factor $\beta_\ell$, we need to understand the physical significance of the phase compensation term. The phase compensation term $\beta_\ell$ accounts for the path difference that is corrected for in the virtual array projection...

The total phase difference $\Delta\phi$ is the sum of these two phase differences:

$$\Delta\phi = \Delta\phi_z + \Delta\phi_{\text{horizontal}} = \frac{2\pi d_\ell^z}{\lambda} + \frac{2\pi\sqrt{(d_\ell^x)^2 + (d_\ell^y)^2}}{\lambda}$$

To correct for this phase difference, we need to add the phase compensation term $\beta_\ell$:

$$\beta_\ell = 2d_\ell^z + \frac{(d_\ell^x)^2 + (d_\ell^y)^2}{2Z_0}$$

Therefore, the correct term that completes the equation is: $\boxed{C}$

---

**Error Analysis:** This example shows mathematically sound reasoning leading to an incorrect final selection. The model provides correct physical interpretation and mathematical derivations but selects the wrong multiple-choice option. This suggests challenges in mapping derived expressions to provided answer choices rather than fundamental understanding failures.

