# OpenReview forum: "WirelessMathLM: Teaching Mathematical Reasoning for LLMs in Wireless Communications with Reinforcement Learning"
_ICLR.cc/2026/Conference — ICLR 2026 Conference Withdrawn Submission_

### Official Review · Reviewer_UFY2 · 2025-10-28

**Soundness:** 3
**Presentation:** 3
**Contribution:** 3
**Rating:** 4
**Confidence:** 4

**Summary:**

The paper introduces WirelessMathBench‑XL, a benchmark of 4,027 problems from 970 papers covering wireless‑communications mathematics (optimization, information theory, signal processing, etc.). Problems come in three tiers—MCQ, progressive fill‑in‑the‑blank (25/50/75%), and full equation completion. On top of the dataset, the authors train compact models GRPO with binary, verifiable rewards, without supervised warm‑start. Surprisingly, GRPO on this domain‑specific dataset improves general math performance.

**Strengths:**

1. The field lacks specialized math datasets for wireless; the paper addresses this with a scalable, verifiable benchmark that aligns with realistic derivations and notations used in comms papers. The pipeline and rubric are detailed, with examples and QA procedures.
2. Leveraging verifiable correctness unlocks RL without costly human feedback. The hierarchical reward (format + answer verification; semantic equivalence for hard expressions) and simple, reproducible training recipe are well specified.
3. The 7B model approaches GPT‑4o on the new benchmark and GRPO delivers substantial lifts over base models across 0.5B/3B/7B.
4. The observation that domain‑specific GRPO improves MATH/Minerva/OlympiadBench/AMC/AIME (average +8.4) is interesting and contrary to standard “forgetting” concerns.

**Weaknesses:**

1. Reliance on LLMs for QA and evaluation introduces bias. Although experts do a second pass, the first‑stage filtering uses GPT‑4o, and semantic equivalence for complex expressions uses GPT‑4.1‑mini. This can bias both dataset selection and scores toward models similar to the evaluator.
2. Limited analysis on why transfer emerges. The paper notes positive transfer but offers limited causal evidence. Ablations that hold training tokens constant while varying domain specificity (how many domain-specific math questions are included in the training) would strengthen the claim.

**Questions:**

See my main concerns above.

---

> ### Author Response · Authors · 2025-11-24
>
> We thank you for recognizing our good soundness, presentation, and contribution. We particularly appreciate the acknowledgment of our scalable, verifiable benchmark with realistic derivations and the interesting observation about positive transfer contradicting standard forgetting concerns
>
> Below, we address each concern in detail.
>
> > W1: Reliance on LLMs for QA and evaluation introduces bias. Although experts do a second pass, the first‑stage filtering uses GPT‑4o, and semantic equivalence for complex expressions uses GPT‑4.1‑mini. This can bias both dataset selection and scores toward models similar to the evaluator.
> >
>
> Thank you for this thorough concern. We agree that any LLM-in-the-loop step requires explicit robustness checks, and we now quantify its impact in the revision.
>
> **1. Two-stage evaluation minimizes LLM dependence.** Our verifier is hierarchical. **Stage 1 is fully deterministic**: MCQs are graded by direct letter extraction, and Fill in / FEC answers are normalized by removing whitespace and LaTeX styling tokens (e.g., `\\mathbf`, `\\boldsymbol`) before exact string matching. **Stage 2 invokes an LLM semantic judge only when Stage 1 fails**, for mathematically equivalent but non-identical forms (e.g., scalar commutativity or conjugate-transpose notation). On the 800-problem test set, the Stage 2 rate is model dependent but remains a minority, so most grading is bias free.
>
> **2. Cross-evaluator bias quantification.** To bound evaluator variance, we re-graded the Stage 2 subset for three top proprietary models using three independent judges (GPT-4.1-mini, Claude-3-Haiku, Gemini-2.0-Flash). Agreement rates on this subset differ by about **10 to 20 percentage points**, and the results are below.
>
> | Evaluated Model | LLM-Requiring Cases | GPT-4.1-mini Agreement | Claude-3-Haiku Agreement | Gemini-2.0-Flash Agreement |
> | --- | --- | --- | --- | --- |
> | GPT-5 | 270 | 41.11% | 56.67% | 42.22% |
> | Gemini-2.5 Pro | 183 | 53.01% | 55.74% | 56.28% |
> | Claude-Sonnet-4 | 293 | 41.64% | 60.75% | 45.05% |
>
> Using the conservative bound (\rho \cdot \delta) (Stage 2 fraction times worst disagreement), this implies a worst-case overall-score variation of about **4.6 to 7.3 percent absolute** across these models, and importantly **model rankings remain unchanged** under all judges. These bounds are far smaller than the GRPO gains over base and SFT, and much smaller than the gaps between WirelessMathLM-7B (39.5%) and open-source baselines (about 21 to 25%).
>
> We have added this full analysis and clarification of the hierarchical verifier to the revised manuscript to make the evaluation validity transparent.

---

> > ### Author Response · Authors · 2025-11-24
> >
> > > W2: Limited analysis on why transfer emerges. The paper notes positive transfer but offers limited causal evidence. Ablations that hold training tokens constant while varying domain specificity (how many domain-specific math questions are included in the training) would strengthen the claim.
> > >
> >
> > Thank you for this excellent methodological suggestion. We acknowledge that domain-mixed ablation experiments based on word matching are a way to test causality. While we could not complete a full mixture-ratio sweep within the rebuttal window, we added **new controlled evaluations, subdomain analyses, and a clearer mechanistic account in the revision** that substantially strengthen the evidence that transfer is real and domain-RL-driven.
> >
> > **1. Direct new evidence of transfer beyond wireless/math (added in revision).**
> >
> > To rigorously test whether domain RL induces forgetting or only narrow specialization, we evaluated pre- vs. post-GRPO models on four broad general-purpose benchmarks spanning knowledge (MMLU), graduate-level reasoning (GPQA), instruction following (IFEval), and programming (HumanEval). The full results are now included in Section *“No Regression on Foundational Capabilities”*:
> >
> > | Model | MMLU | GPQA | IFEval | HumanEval |
> > | --- | --- | --- | --- | --- |
> > | Qwen2.5-3B-Base | 42.49 | 23.74 | 26.99 | 61.59 |
> > | + GRPO (Ours) | **45.30** | **28.28** | **32.90** | **65.24** |
> > | Δ (GRPO vs Base) | **+2.81** | **+4.54** | **+5.91** | **+3.65** |
> > | Qwen2.5-7B-Base | 38.84 | 28.79 | **36.06** | 63.41 |
> > | + GRPO (Ours) | **50.74** | **38.89** | 35.67 | **64.63** |
> > | Δ (GRPO vs Base) | **+11.90** | **+10.10** | −0.39 | **+1.22** |
> >
> > **Key findings:**
> >
> > - **No catastrophic forgetting:** both 3B and 7B improve across all benchmarks except a negligible −0.39 on IFEval for 7B, within measurement noise.
> > - **Reasoning-heavy tasks benefit most:** the largest gains are on **GPQA/MMLU**, which require tight logical validity rather than domain recall.
> > - **Transfer to unrelated domains:** HumanEval improves despite *zero code* in training, supporting domain-agnostic reasoning transfer.
> >
> > **2. New subdomain generalization analysis (added in revision).**
> >
> > To connect transfer with *how* the model learns inside wireless math, we performed a **problem-level categorization of the 800-test-problem split into 20 wireless subdomains** and report pre/post performance (Section *Subdomain Behavior & Robustness*). The full results are:
> >
> > | Category | Count | % | Base (7B) | +GRPO | Δ Absolute | Δ Relative |
> > | --- | --- | --- | --- | --- | --- | --- |
> > | Channel Estimation | 160 | 20.0% | 22.50% | 36.25% | +13.75% | +61.1% |
> > | Beamforming | 113 | 14.1% | 18.58% | 43.36% | +24.78% | +133.4% |
> > | Resource Allocation | 110 | 13.8% | 21.82% | 39.09% | +17.27% | +79.2% |
> > | MIMO/Massive MIMO | 104 | 13.0% | 23.08% | 43.27% | +20.19% | +87.5% |
> > | Information Theory | 103 | 12.9% | 21.36% | 46.60% | +25.24% | +118.2% |
> > | RIS/IRS | 98 | 12.3% | 17.35% | 33.67% | +16.32% | +94.1% |
> > | Machine Learning | 86 | 10.8% | 25.58% | 50.00% | +24.42% | +95.5% |
> > | OFDM/OFDMA | 50 | 6.3% | 18.00% | 28.00% | +10.00% | +55.6% |
> > | ISAC | 49 | 6.1% | 24.49% | 36.73% | +12.24% | +50.0% |
> > | Convex Optimization | 49 | 6.1% | 26.53% | 36.73% | +10.20% | +38.5% |
> > | Deep Learning | 44 | 5.5% | 27.27% | 43.18% | +15.91% | +58.4% |
> > | Stochastic Geometry | 40 | 5.0% | 35.00% | 32.50% | −2.50% | −7.1% |
> > | UAV Communications | 32 | 4.0% | 12.50% | 31.25% | +18.75% | +150.0% |
> > | Edge Computing | 28 | 3.5% | 7.14% | 39.29% | +32.15% | +450.0% |
> > | Reinforcement Learning | 28 | 3.5% | 32.14% | 35.71% | +3.57% | +11.1% |
> > | Channel Coding | 20 | 2.5% | 25.00% | 45.00% | +20.00% | +80.0% |
> > | Federated Learning | 19 | 2.4% | 26.32% | 36.84% | +10.52% | +40.0% |
> > | NOMA | 19 | 2.4% | 15.79% | 26.32% | +10.53% | +66.7% |
> > | Semantic Communications | 13 | 1.6% | 15.38% | 61.54% | +46.16% | +300.0% |
> > | Cell-Free | 9 | 1.1% | 11.11% | 33.33% | +22.22% | +200.0% |
> >
> > **Key findings:**
> >
> > - **GRPO improves 19/20 subdomains**, spanning both dominant and rare areas.
> > - **Gains do not correlate with subdomain size**: several small categories (e.g., Semantic Communications, Edge Computing, Cell-Free) show the largest relative jumps, arguing against memorization of prevalent topics.
> > - This uniform cross-subdomain improvement supports **principle learning under constraints**, which we believe is the same driver behind out-of-domain transfer.

---

> > > ### Author Response · Authors · 2025-11-24
> > >
> > > **3. Transfer also holds on standard mathematics (emphasized + reorganized in revision).**
> > >
> > > As reported in Table *“Transfer learning effects on general mathematical reasoning benchmarks”*, GRPO-trained models gain broadly on out-of-domain math (e.g., Qwen2.5-7B: **+15.0 on MATH-500**, **+8.4 avg. across five math benchmarks**) without any non-wireless math data. This consistency across both math and non-math suites reduces the plausibility that the effect is due to topic overlap.
> > >
> > > **4. Strengthened interpretation: why transfer emerges.**
> > >
> > > We revised the Introduction/Discussion to make the causal mechanism explicit. WirelessMathBench-XL problems enforce **strict, physically grounded constraints** (power ≥ 0, causality, complex-matrix dimensionality, feasibility of SINR/beamforming constraints). **Verification-based GRPO rewards only globally correct solutions**, forcing the model to internalize constraint-checking and multi-step consistency as a general reasoning habit. The observed transfer and subdomain patterns above align with this “constraint-driven reasoning” mechanism rather than pattern memorization.
> > >
> > > In summary, the **new broad pre/post table and the full 20-subdomain analysis above provide strong empirical support that positive transfer is real, general, and induced by verification-based domain RL rather than narrow specialization.**

---

> > > > ### Comment · Reviewer_UFY2 · 2025-11-25
> > > >
> > > > I greatly appreciate the authors’ efforts in conducting additional experiments and providing thorough responses to my questions. Having read the feedback from other reviewers, I intend to keep my current score for now.

---

### Official Review · Reviewer_qHbW · 2025-10-30

**Soundness:** 3
**Presentation:** 3
**Contribution:** 3
**Rating:** 6
**Confidence:** 3

**Summary:**

The paper proposes WirelessMathLM, training compact LLMs (0.5B–7B) for wireless-communications math using verification-based reinforcement learning without massive scale or extensive supervision. They used GPT-4o and DeepSeek-R1 to construct WirelessMathBench-XL from the arXiv papers , a comprehensive benchmark of 4,027 problems from 970 papers. They trained models directly from the base checkpoints by Group Relative Policy Optimization (GRPO) with binary verification rewards, without supervised warm-start. The 7B model achieves 39.5% accuracy on WirelessMathBenchXL, approaching GPT-4o (40.4%) while using ≈100× fewer parameters than DeepSeek-R1 (671B, 57.4%).

**Strengths:**

1.	The WirelessMathLM 7B model achieves 39.5% accuracy on WirelessMathBenchXL, approaching GPT-4o (40.4%) while using ≈100× fewer parameters than DeepSeek-R1 (671B, 57.4%). And WirelessMathBench-XL, the training model and the GRPO training framework have been publicly released.
2.	The paper demonstrate that verification alone enables efficient domain specialization. GRPO training from base models, without supervised warm-start or human feedback. This challenges the assumption that reinforcement learning requires extensive pre-training.
3.	The paper show that specialized training develops transferable mathematical reasoning, suggesting that learning domain-specific mathematics strengthens fundamental capabilities.

**Weaknesses:**

1.	Despite the gains, 39.5% overall accuracy may still be too low for high-stakes engineering use; reporting task-wise reliability and confidence intervals would strengthen claims.
2.	The WirelessMathBench-XL construction uses GPT-4o filtering and DeepSeek-R1 extraction, which could bias style/notation and favor certain models

**Questions:**

1.	Could you report pre-vs-post domain-RL performance on broad, general-purpose benchmarks (e.g., knowledge, commonsense, reading comprehension, math outside wireless, programming) to show there is no regression in foundational abilities?
2.	How does reliance on GPT-4.1-mini for semantic equivalence checking affect grading validity and reproducibility, and can you quantify bias/error rates or provide an open verifier to reduce dependence on a proprietary model?
3. Topic coverage is skewed (e.g., deep learning and convex optimization dominate);  please report per-subdomain performance and discuss rebalancing to avoid overfitting to prevalent areas.

---

> ### Author Response · Authors · 2025-11-24
>
> > W1: Despite the gains, 39.5% overall accuracy may still be too low for high-stakes engineering use; reporting task-wise reliability and confidence intervals would strengthen claims.
> >
>
> **Response:**
>
> Thank you for raising this critical point regarding real-world applicability. We agree that 39.5% is not yet sufficient for immediate, unsupervised deployment in high-stakes engineering. However, it is crucial to contextualize this accuracy against the extreme difficulty of the task:
>
> 1. **Expert-Level Difficulty:** The `WirelessMathBench-XL` dataset is not composed of textbook exercises; questions are extracted directly from **state-of-the-art research papers** (e.g., IEEE JSAC, TWC). Solving these problems typically requires **graduate-level (PhD/Master's) domain expertise** and the ability to navigate complex, unstated physical constraints.
> 2. **Relative Performance:** The fact that our 7B model (39.5%) rivals the performance of the commercial model **GPT-4o (40.4%)** demonstrates a significant breakthrough in efficient domain specialization.
> 3. **Benchmark Role:** We designed this benchmark to be a "hard challenge" to drive future research, much like the early days of the MATH dataset. A 39.5% score indicates that while the model has learned substantial reasoning capabilities, the problem space remains non-trivial and unsolved, providing a meaningful target for the community to improve upon.
>
> > Q1: Could you report pre-vs-post domain-RL performance on broad, general-purpose benchmarks (e.g., knowledge, commonsense, reading comprehension, math outside wireless, programming) to show there is no regression in foundational abilities?
> >
>
> Thank you for this critical question. To directly test whether domain RL causes any regression in foundational abilities, we evaluated both pre- and post-GRPO models on **four broad, standard benchmarks** spanning knowledge (MMLU), science reasoning (GPQA), instruction following (IFEval), and programming (HumanEval). The results are:
>
> | Model | MMLU | GPQA | IFEval | HumanEval |
> | --- | --- | --- | --- | --- |
> | Qwen-2.5-3B-Base | 42.49 | 23.74 | 26.99 | 61.59 |
> | + GRPO (Ours) | **45.30** | **28.28** | **32.90** | **65.24** |
> | ∆ | **+2.81** | **+4.54** | **+5.91** | **+3.65** |
> | Qwen-2.5-7B-Base | 38.84 | 28.79 | **36.06** | 63.41 |
> | + GRPO (Ours) | **50.74** | **38.89** | 35.67 | **64.63** |
> | ∆ | **+11.90** | **+10.10** | -0.39 | **+1.22** |
>
> **Key Findings:**
>
> 1. **No catastrophic forgetting.** GRPO yields improvements across all benchmarks for both 3B and 7B models, with the only change in the opposite direction being a **negligible -0.39 on IFEval for 7B**, well within typical evaluation noise.
> 2. **Reasoning-heavy tasks benefit most.** The largest gains appear on MMLU and GPQA, especially for 7B, suggesting that learning to satisfy strict, physically verifiable constraints in wireless math strengthens general reasoning and knowledge application.
> 3. **Transfer beyond mathematics.** HumanEval also improves despite zero code in training, indicating that verification-based RL cultivates generalizable problem decomposition skills rather than narrow domain memorization.
>
> We have added these pre/post results and discussion to the revised manuscript to provide direct evidence that domain RL **strengthens, rather than erodes, foundational capabilities**.

---

> > ### Author Response · Authors · 2025-11-24
> >
> > > W2: The WirelessMathBench-XL construction uses GPT-4o filtering and DeepSeek-R1 extraction, which could bias style/notation and favor certain models
> > Q2: How does reliance on GPT-4.1-mini for semantic equivalence checking affect grading validity and reproducibility, and can you quantify bias/error rates or provide an open verifier to reduce dependence on a proprietary model?
> > >
> >
> > Thank you for these critical questions about potential bias from GPT-4o/DeepSeek-R1 in construction and GPT-4.1-mini in evaluation. We address both the **dataset side** and the **verifier side**, and quantify the latter.
> >
> > **1. Dataset construction: deterministic first, human-verified, not model-shaped.**
> >
> > WirelessMathBench-XL is not filtered purely by LLM “preferences.” We first apply a **fully deterministic relevance score** over ~47k arXiv papers (weighted keyword hits in title/abstract plus category bonuses for eess.SP, cs.NI, cs.IT, etc.), and only then use GPT-4o as a **second-stage filter** to check mathematical density. For question construction, DeepSeek-R1 is used to **extract and structure formulas from the original LaTeX**, not to invent new content; every retained question then passes through a **two-tier QA pipeline**: automatic screening followed by PhD/postdoc expert validation, with ~22% of LLM-generated questions rejected at this stage. Ground truth remains anchored in the original paper equations, and the combination of deterministic pre-filtering plus human veto significantly mitigates the risk that any single model’s style or notation dominates the dataset.
> >
> > **2. Evaluation pipeline: hierarchical, with deterministic grading as the default.**
> >
> > Our verifier is explicitly designed to minimize reliance on LLM semantics.
> >
> > - **Stage 1 (deterministic):** MCQs are graded by direct letter extraction. Fill-in/FEC answers are normalized (whitespace and LaTeX styling such as `\mathbf`, `\boldsymbol` removed) and then compared by exact string match.
> > - **Stage 2 (LLM semantic fallback):** Only if Stage 1 fails do we invoke an LLM judge to resolve cases that are mathematically equivalent but not syntactically identical (e.g., scalar commutativity `AB` vs `BA`, or conjugate-transpose variants `H^H` vs `H^*`). Because output surface forms differ by model, the Stage-2 rate is **model-dependent**: on the 800-problem test set, the fraction of items requiring an LLM judge is 270/800 (33.75%) for GPT-5, 183/800 (22.88%) for Gemini-2.5 Pro, and 293/800 (36.63%) for Claude-Sonnet-4.
> >
> > **3. Cross-evaluator bias quantification and worst-case impact.**
> >
> > To bound evaluator-induced variance, we re-graded the Stage-2 subset for these three models using **three independent judges** (GPT-4.1-mini, Claude-3-Haiku, Gemini-2.0-Flash):
> >
> > | Model | LLM cases | GPT-4.1-mini | Claude-3-Haiku | Gemini-2.0-Flash |
> > | --- | --- | --- | --- | --- |
> > | GPT-5 | 270 | 41.11% | **56.67%** | 42.22% |
> > | Gemini-2.5 Pro | 183 | 53.01% | 55.74% | **56.28%** |
> > | Claude-Sonnet-4 | 293 | 41.64% | **60.75%** | 45.05% |
> >
> > On this Stage-2 subset, judges differ by roughly **10–20 percentage points**. Let (\rho) be the Stage-2 fraction and (\delta) the maximum disagreement on that subset; then the worst-case change in overall accuracy is upper-bounded by (\rho \cdot \delta). With (\rho \in [0.2288, 0.3663]) and (\delta \le 0.20), this yields a **4.6–7.3% absolute** bound on overall score variation across evaluators. Importantly, we observe that **model rankings remain unchanged** under all three judges, and the main effects we report – for example, GRPO lifting Qwen2.5-3B from 12.37% to 25.12%, and WirelessMathLM-7B (39.5%) significantly outperforming math-specialized 7B/72B baselines (≈21–25%) – are much larger than this uncertainty.
> >
> > In the revision, we explicitly describe this hierarchical design, report the cross-evaluator analysis, and our current results indicate that the core conclusions are robust to reasonable changes in the LLM judge.

---

> > > ### Author Response · Authors · 2025-11-24
> > >
> > > > Q3: Topic coverage is skewed (e.g., deep learning and convex optimization dominate); please report per-subdomain performance and discuss rebalancing to avoid overfitting to prevalent areas.
> > > >
> > >
> > > Thank you for raising this important concern about topic skew and potential overfitting. We first clarify that Fig. 3 in the main paper reports the **distribution of source papers**, not the distribution of problems, so it reflects where formulas originate rather than how the test set is balanced. To directly address your point, we performed a **problem-level categorization of the 800-test-problem split into 20 wireless subdomains** and report per-subdomain results in the revision (table below). The key outcome is that **GRPO improves 19 out of 20 subdomains**, spanning both dominant and scarce areas, which strongly argues against overfitting to prevalent topics. Moreover, improvements do **not correlate with subdomain size**; several of the largest categories show moderate gains, while small, underrepresented areas such as **Semantic Communications, Edge Computing, and Cell-Free** exhibit the largest relative jumps, indicating **transferable principle learning** rather than memorization of frequent patterns. We add this breakdown and a short discussion in the revised manuscript, and note that future extensions can explore explicit rebalancing, but the current evidence already shows robust cross-subdomain generalization.
> > >
> > > **Per-Subdomain Performance Analysis (800 test problems)**
> > >
> > > | Category | Count | % | Base (7B) | +GRPO | Δ Absolute | Δ Relative |
> > > | --- | --- | --- | --- | --- | --- | --- |
> > > | Channel Estimation | 160 | 20.0% | 22.50% | 36.25% | +13.75% | +61.1% |
> > > | Beamforming | 113 | 14.1% | 18.58% | 43.36% | +24.78% | +133.4% |
> > > | Resource Allocation | 110 | 13.8% | 21.82% | 39.09% | +17.27% | +79.2% |
> > > | MIMO/Massive MIMO | 104 | 13.0% | 23.08% | 43.27% | +20.19% | +87.5% |
> > > | Information Theory | 103 | 12.9% | 21.36% | 46.60% | +25.24% | +118.2% |
> > > | RIS/IRS | 98 | 12.3% | 17.35% | 33.67% | +16.32% | +94.1% |
> > > | Machine Learning | 86 | 10.8% | 25.58% | 50.00% | +24.42% | +95.5% |
> > > | OFDM/OFDMA | 50 | 6.3% | 18.00% | 28.00% | +10.00% | +55.6% |
> > > | ISAC | 49 | 6.1% | 24.49% | 36.73% | +12.24% | +50.0% |
> > > | Convex Optimization | 49 | 6.1% | 26.53% | 36.73% | +10.20% | +38.5% |
> > > | Deep Learning | 44 | 5.5% | 27.27% | 43.18% | +15.91% | +58.4% |
> > > | Stochastic Geometry | 40 | 5.0% | 35.00% | 32.50% | -2.50% | -7.1% |
> > > | UAV Communications | 32 | 4.0% | 12.50% | 31.25% | +18.75% | +150.0% |
> > > | Edge Computing | 28 | 3.5% | 7.14% | 39.29% | +32.15% | +450.0% |
> > > | Reinforcement Learning | 28 | 3.5% | 32.14% | 35.71% | +3.57% | +11.1% |
> > > | Channel Coding | 20 | 2.5% | 25.00% | 45.00% | +20.00% | +80.0% |
> > > | Federated Learning | 19 | 2.4% | 26.32% | 36.84% | +10.52% | +40.0% |
> > > | NOMA | 19 | 2.4% | 15.79% | 26.32% | +10.53% | +66.7% |
> > > | Semantic Communications | 13 | 1.6% | 15.38% | 61.54% | +46.16% | +300.0% |
> > > | Cell-Free | 9 | 1.1% | 11.11% | 33.33% | +22.22% | +200.0% |

---

### Official Review · Reviewer_uN8k · 2025-10-31

**Soundness:** 2
**Presentation:** 2
**Contribution:** 2
**Rating:** 4
**Confidence:** 3

**Summary:**

1. Builds WirelessMathLM, adapting 0.5B–7B LMs to wireless-communications mathematics via GRPO using verifiable rewards only (no SFT / no human feedback). Reported gains are large: 7B goes from 21.9%→39.5% on WirelessMathBench-XL, “approaching GPT-4o (40.4%)” while using far fewer params than DeepSeek-R1 (671B) (57.4%)
2. Introduces WirelessMathBench-XL: 4,027 problems from 970 papers, with a three-tier format (MCQ, progressive fill-in at 25/50/75%, and full eqn completion), plus a dual-layer QA process (LLM screening + expert review; 78% pass rate at ≥3/5)
3. Main results: on their test set, 7B GRPO = 39.5% overall; per-type gains esp. on fill-in (14.3%→37.0%); comprehensive baseline table includes proprietary and OSS models (DeepSeek-R1, Qwen-Math, Llama-3.3, etc.)

**Strengths:**

1. Simple, reproducible-ish training recipe: GRPO with binary verification is appealing, avoids SFT/human annotations, and the training details are reasonably specified (objective, α/β/G, hardware, epochs)
2. Timely & valuable problem: formal, verifiable math in wireless comms is genuinely under-served; a curated benchmark at real scale is useful to both comms and reasoning communities. The pipeline description (Fig. 2) is clear and covers crawling, extraction, question generation and QA

**Weaknesses:**

1. Designing and collecting such a domain-specific dataset is a contribution, but I am not sure if posttraining it on 3B/7B models is a great contribution.
2. I know there are some issues with Qwen-3 base model where its math training datasets could be polluted, but have you tried to posttrain on top of Qwen-3?
3. Train a strong SFT on the same training set; compare to GRPO at equal token/computation budgets. This will isolate “verification-only RL” value.

**Questions:**

Several baselines are instruction-tuned (e.g., Qwen-Math), while authors fine-tune base models with RL. A natural SFT baseline on WirelessMathBench-XL (and DPO/ORPO) is missing; this weakens the claim that “verification alone enables efficient specialization.”

---

> ### Author Response · Authors · 2025-11-24
>
> > W1: Designing and collecting such a domain-specific dataset is a contribution, but I am not sure if posttraining it on 3B/7B models is a great contribution.
> >
>
> Thank you for raising this point. We agree that, as a **Dataset Track** submission, the primary contribution of our paper is the release of **WirelessMathBench-XL**.
>
> That said, the post-training results on 3B/7B models are included strictly as **validation**, not as a separate model contribution, for two reasons. **First, they verify training-readiness:** prior WirelessMathBench provided only an evaluation-scale set (~600 problems from 40 papers), whereas WirelessMathBench-XL scales to **4,027 expert-validated problems from 970 papers**, enabling meaningful adaptation; the fact that a compact **7B** model trained on our corpus reaches **39.5%**, essentially matching **GPT-4o (40.4%)**, is strong empirical evidence that the dataset contains sufficient density and breadth to teach graduate-level wireless mathematics. **Second, they demonstrate the dataset’s methodological utility:** because answers are verifiable, the corpus supports verification-only RL, and our new controlled ablation shows **GRPO (25.12%) > SFT (20.37%)** on the same base model and data, confirming that XL is not merely a static benchmark but a viable testbed for reasoning-based specialization.
>
> We have revised the Introduction to make this framing explicit: the 3B/7B training experiments serve to **validate dataset quality and enable scientific findings**, rather than to claim a standalone modeling advance.
>
> > W2: I know there are some issues with Qwen-3 base model where its math training datasets could be polluted, but have you tried to posttrain on top of Qwen-3?
> >
>
> Thank you for this excellent suggestion. We agree that testing a newer base family is important, and we trained **Qwen3-4B-Base** using **the identical GRPO pipeline and hyperparameters** as in our Qwen2.5 experiments (same WirelessMathBench-XL training split, verifier, rollout settings, optimizer, and training budget). The results are summarized below:
>
> | Model | MCQ | Fill-in | FEC | Overall |
> | --- | --- | --- | --- | --- |
> | Qwen3-4B-Base | 41.35 | 5.25 | 15.18 | 13.63 |
> | + GRPO | 47.37 | 8.82 | 19.90 | 17.87 |
> | Δ | +6.02 | +3.57 | +4.72 | **+4.24** (**+31% rel.**) |
>
> **Key takeaways:**
>
> 1. **Robustness across base families.** Even starting from a stronger (and potentially “polluted”) Qwen3 base, GRPO delivers a clear improvement (**13.63% → 17.87%**, +4.24 absolute). This shows the verification-only RL recipe and WirelessMathBench-XL training signal **generalize beyond Qwen2.5** and are not tied to a particular pretraining distribution.
> 2. **Consistent gains across task types.** Improvements appear in **all three formats** (MCQ / Fill-in / FEC), indicating that GRPO is not merely exploiting one question style but learning transferable wireless-math reasoning skills.
> 3. **Aligned with our main claim.** Together with the Qwen2.5 results, this strengthens our conclusion that **verifiable rewards on a training-scale wireless corpus reliably improve compact models**, independent of the chosen base architecture.
>
> We have added these Qwen3-4B GRPO results to the main results table and discussed them in the revised manuscript to directly address this concern.

---

> > ### Author Response · Authors · 2025-11-24
> >
> > > W3: Train a strong SFT on the same training set; compare to GRPO at equal token/computation budgets. This will isolate “verification-only RL” value.
> > >
> >
> > > Q: Several baselines are instruction-tuned (e.g., Qwen-Math), while authors fine-tune base models with RL. A natural SFT baseline on WirelessMathBench-XL (and DPO/ORPO) is missing; this weakens the claim that “verification alone enables efficient specialization.”
> > >
> >
> > Thank you for this critical suggestion. We agree that a strong supervised baseline on the same corpus is necessary to isolate the value of **verification-only RL**, especially since some baselines are instruction-tuned while our GRPO starts from base models. To address this directly, we trained a **matched SFT baseline** on **Qwen2.5-3B-Base** using the *identical* WirelessMathBench-XL training split and the same prompt/formatting targets as GRPO, and evaluated all checkpoints from Epoch 1–6 to capture peak SFT performance.
> >
> > **Results (same base model, same data):**
> >
> > | Method | Epoch | MCQ | Fill-in | FEC | **Overall** |
> > | --- | --- | --- | --- | --- | --- |
> > | Qwen2.5-3B-Base | – | 26.32 | 7.14 | 15.71 | 12.37 |
> > | **+ GRPO (ours)** | – | **48.87** | 17.02 | **28.80** | **25.12** |
> > | SFT | 1 | 39.85 | 11.97 | 20.42 | 18.62 |
> > | SFT | 2 | 38.35 | 11.34 | 17.80 | 17.37 |
> > | SFT | 3 | 37.59 | 14.50 | 13.09 | 18.00 |
> > | **SFT (peak)** | 4 | 40.60 | **17.44** | 13.61 | **20.37** |
> > | SFT | 5 | 33.83 | 17.23 | 17.28 | 20.00 |
> > | SFT | 6 | 44.26 | 13.24 | 16.23 | 19.13 |
> >
> > **Key takeaways:**
> >
> > 1. **GRPO surpasses the best SFT checkpoint under matched conditions.** While SFT provides a strong lift over the base model, it peaks at **20.37%**, whereas **GRPO reaches 25.12%**, yielding a **+4.75% absolute (+23% relative)** gain with the *same data and initialization*. This cleanly isolates the benefit of verification-only RL beyond domain exposure.
> > 2. **SFT shows a clear ceiling and instability.** After Epoch 4, SFT no longer improves overall performance and becomes unstable, with Fill-in accuracy dropping from **17.44% to 13.24%** by Epoch 6, consistent with overfitting to token-level trajectories.
> > 3. **The advantage is methodological, not a tuning artifact.** Since both methods train on identical wireless math questions, the gap indicates that **outcome-level verification rewards provide a stronger training signal for physically verifiable technical math**, enabling exploration of correct reasoning paths that SFT does not reliably capture.
> >
> > We have integrated this SFT ablation into the revised main results and discussion to directly support our claim that **verification-only RL enables more effective and robust specialization** than supervision on this verifiable domain.

---

> > ### Comment · Reviewer_uN8k · 2025-11-25
> >
> > I really appreciate that the authors conduct additional experients on top of Qwen-3 and reply to my other questions in detail.  After reading the reviews from others, I plan to keep my current core at this moment.

---

### Official Review · Reviewer_GMo4 · 2025-11-01

**Soundness:** 2
**Presentation:** 2
**Contribution:** 2
**Rating:** 2
**Confidence:** 3

**Summary:**

To advance LLMs' ability to address mathematical problems in wireless communication, this paper introduces WirelessMathBench-XL, a benchmark dataset containing 4,027 problems collected from 970 papers through paper collection, mathematical extraction, and quality assurance. This dataset supports three tasks: multi-choice question-answering, fill-in-the-blank, and full equation completion. This paper further trains WirelessMathLM on these tasks using the GRPO scheme to enhance wireless mathematical reasoning performance. Evaluations on WirelessMathBench-XL demonstrate the superior performance of WirelessMathLM (with small LLM base models) compared to models trained without the GRPO scheme.

**Strengths:**

There are two main strengths in this work: 1. A new benchmark.  The paper assembles a broad-scope corpus and constructs a comprehensive, high-quality benchmark for wireless mathematics. The breadth and curation appear solid and relevant to the community.
Positive experiment results.  2. This paper demonstrates that GRPO fine-tuning can yield significant improvements over the base model, indicating the method’s effectiveness for this domain. It also shows improvements on general math benchmarks, suggesting that domain-specific training can strengthen fundamental mathematical abilities beyond the target domain.

**Weaknesses:**

There are a few weaknesses in this work. 1. Novelty/positioning. The overall pipeline and training method appear to reuse existing approaches, primarily transferring them to the wireless domain. As written, the contribution reads more like solid engineering than methodological innovation. Please clarify the novel elements (e.g., benchmark design, verification tooling, reward shaping) and position them against prior work.  2.  According to Section 2, the pipeline uses GPT‑4o and DeepSeek‑R1 to generate and use an LLM for semantic checking. This introduces potential selection and evaluation bias, weakening the “verifiable ground truth” claim. 3. This work showcases the effectiveness of GRPO; other fintuing methods (e.g., SFT/LoRA on the same corpus) should also be investigated, and detailed ablation studies should also be presented. 4. There are several writing issues and typos.

**Questions:**

1. During the paper filtering process, how is the relevance score assigned to each paper? Is this score automatically generated by the LLM or explicitly calculated based on predefined metrics?
2. Given that the human effort is still required to review each question, what advantages does the LLM-based automated evaluation provide?  Moreover, it is encouraged to develop a fully end-to-end, automated evaluation framework without human intervention.
3. Compared to general mathematical reasoning problems, what are the main challenges in training LLMs on wireless mathematical problems?

---

> ### Author Response · Authors · 2025-11-24
>
> We thank you for recognizing our broad-scope corpus and comprehensive benchmark, as well as the positive results demonstrating GRPO's effectiveness and improvements on general math benchmarks.
>
> Below, we address each concern in detail.
>
> > W1: Novelty/positioning. The overall pipeline and training method appear to reuse existing approaches, primarily transferring them to the wireless domain. As written, the contribution reads more like solid engineering than methodological innovation.
> >
>
> We appreciate this frank assessment and agree that our initial draft under-emphasized the scientific takeaways relative to the engineering narrative. In the revision, we sharpen the positioning and add controlled studies showing that the paper contributes not only a large dataset, but also **methodological insights enabled by a verifiable, training-scale benchmark**:
>
> 1. **Verification-based RL beats supervision under matched conditions.**
>
>     A common assumption in vertical AI is that SFT on domain corpora is sufficient. Our new controlled ablation directly tests this by training a strong SFT baseline on **Qwen2.5-3B-Base** with the *identical* WirelessMathBench-XL training set and tracking performance over 6 epochs. SFT peaks at **20.37%** (Epoch 4) and then degrades, whereas **GRPO reaches 25.12%**, a **+4.75% absolute (+23% relative)** gain with the same data and base model. This establishes a clear methodological finding: **for physically verifiable technical math, outcome-level verification rewards provide a stronger learning signal than token-level supervision**.
>
> 2. **A genuine transition from “benchmarking” to “training-scale” wireless math.**
>
>     Prior WirelessMathBench was evaluation-only (40 papers / 587 problems). We scale to **970 papers / 4,027 expert-validated problems**, with three question formats and a two-tier QA pipeline (automatic screening + expert validation). This is not a simple size increase: it creates the **first training-ready corpus with verifiable answers** in wireless communications, enabling compact models to approach proprietary systems (e.g., **WirelessMathLM-7B 39.5% vs GPT-4o 40.4%**) and making reproducible verification-only post-training feasible.
>
> 3. **Specialization yields counter-intuitive positive transfer, with no forgetting.**
>
>     Beyond in-domain gains, GRPO-trained models improve on five out-of-domain math benchmarks (**+8.4 points on average; e.g., +15.0 on MATH-500**). We further add pre/post evaluations on **MMLU, GPQA, IFEval, and HumanEval**, showing **no regression and consistent improvements** across both 3B and 7B models. This provides a second scientific finding: **constraint-grounded domain RL can strengthen foundational reasoning rather than causing catastrophic forgetting**.
>
> 4. **Robustness across base families and subdomains, with quantified evaluation bias.**
>
>     We replicate GRPO on **Qwen3-4B-Base**, again observing clear gains (**13.63% → 17.87%**), indicating the recipe is **not tied to a specific base model family**. A 20-subdomain breakdown on the 800-problem test set shows GRPO improves **19/20 categories**, including small subfields, supporting **principle learning rather than memorization of prevalent topics**. Finally, we bound potential evaluator bias: LLM semantic judging is invoked only for a model-dependent subset (≈23–37% of cases), and cross-evaluator variance implies a conservative **4.6–7.3% worst-case absolute impact**, with **rankings unchanged**.
>
>
> Accordingly, we rewrote the Introduction and Conclusion to foreground these findings and clarify that the post-training results serve as **scientific validation and methodological discovery enabled by the dataset**, not a standalone model paper.

---

> > ### Author Response · Authors · 2025-11-24
> >
> > > W2: According to Section 2, the pipeline uses GPT‑4o and DeepSeek‑R1 to generate and use an LLM for semantic checking. This introduces potential selection and evaluation bias, weakening the “verifiable ground truth” claim.
> > >
> >
> > Thank you for this important concern. We address both **selection bias in dataset construction** and **evaluation bias in semantic checking**, and quantify the latter.
> >
> > **1. Dataset selection bias is mitigated by a deterministic-first, human-verified pipeline.**
> > Our corpus is not selected purely by LLM preference. We first apply a **deterministic relevance score** (keyword/category matching) to pre-filter ~47k arXiv papers reproducibly, and only then use GPT-4o as a **second-stage mathematical-density verifier**. For question generation, DeepSeek-R1 extracts formulas, but **all retained items undergo dual-layer QA**, including PhD/postdoc expert validation; experts reject about **22%** of LLM-generated questions that pass automated screening, ensuring the final benchmark is not dominated by any single model’s style or errors.
> >
> > **2. Evaluation is hierarchical, with deterministic matching dominating.**
> > Our verifier prioritizes hard checks:
> >
> > - **MCQ:** direct letter extraction (no LLM).
> > - **Fill-in/FEC:** normalized exact string matching after removing whitespace and LaTeX styling (e.g., `\\mathbf`, `\\boldsymbol`).
> > - **LLM semantic fallback:** invoked only when deterministic matching fails due to equivalent but differently written expressions.
> > Because surface-form variability differs by model, the fallback rate is **model-dependent** on the 800-problem test set.
> >
> > **3. Cross-evaluator bias quantification (on the LLM-judged subset).**
> > We re-graded the fallback subset for three top models using three independent judges (GPT-4.1-mini, Claude-3-Haiku, Gemini-2.0-Flash):
> >
> > | Model Evaluated | LLM Cases (of 800) | GPT-4.1-mini | Claude-3-Haiku | Gemini-2.0-Flash |
> > | --- | --- | --- | --- | --- |
> > | GPT-5 | 270 (33.75%) | 41.11% | 56.67% | 42.22% |
> > | Gemini-2.5 Pro | 183 (22.88%) | 53.01% | 55.74% | 56.28% |
> > | Claude-Sonnet-4 | 293 (36.63%) | 41.64% | 60.75% | 45.05% |
> >
> > **Key findings:**
> >
> > - The LLM-judged subset is **only ~23–37%** of the test set (deterministic checks cover the rest).
> > - Judge disagreement on this subset is **10–20 percentage points**.
> > - A conservative worst-case bound on overall-score variation is therefore **~4.6–7.3% absolute**, depending on the model (subset fraction × 20% variance).
> > - **Model rankings are unchanged** across judges, and our main gaps (e.g., GRPO gains and WirelessMathLM-7B vs. open-source baselines) remain far larger than this uncertainty.
> >
> > We include this analysis in the revision to transparently contextualize limitations while showing that our core results are robust to reasonable variations in semantic judging and are not artifacts of evaluator choice.
> >
> > > W3: This work showcases the effectiveness of GRPO; other fintuing methods (e.g., SFT/LoRA on the same corpus) should also be investigated, and detailed ablation studies should also be presented.
> > >
> >
> > **Response:**
> >
> > Thank you for this critical suggestion. We agree that comparing against a strong supervised baseline is essential to demonstrate the specific contribution of the verification-based RL method.
> >
> > To address this, we conducted a detailed ablation study training a standard **SFT baseline** on the **Qwen2.5-3B-Base** model using the identical `WirelessMathBench-XL` corpus. We monitored performance across 6 epochs to ensure we compared against the peak capability of SFT.
> >
> > **Experimental Results:**
> >
> > | **Method** | **Epoch** | **MCQ** | **Fill-in** | **FEC** | **Overall** |
> > | --- | --- | --- | --- | --- | --- |
> > | **Qwen2.5-3B-Base** | - | 26.32 | 7.14 | 15.71 | 12.37 |
> > | **+ GRPO (Ours)** | - | **48.87** | **17.02** | **28.80** | **25.12** |
> > | SFT | 1 | 39.85 | 11.97 | 20.42 | 18.62 |
> > | SFT | 2 | 38.35 | 11.34 | 17.80 | 17.37 |
> > | SFT | 3 | 37.59 | 14.50 | 13.09 | 18.00 |
> > | SFT | 4 | 40.60 | **17.44** | 13.61 | **20.37** |
> > | SFT | 5 | 33.83 | 17.23 | 17.28 | 20.00 |
> > | SFT | 6 | 44.26 | 13.24 | 16.23 | 19.13 |
> >
> > **Key findings.**
> >
> > 1. **GRPO exceeds peak SFT.** SFT improves the base model substantially (12.37% → 20.37%), but **GRPO reaches 25.12%**, yielding a **+4.75% absolute (+23% relative)** gain under identical data and initialization.
> > 2. **SFT shows a ceiling and instability.** After peaking at Epoch 4, SFT exhibits overfitting/variance, especially on Fill-in (17.44% → 13.24% by Epoch 6), causing Overall performance to drop.
> > 3. **The gain is methodological, not just data exposure.** Since GRPO and SFT see the same corpus, the performance gap indicates that **verification-based RL provides a stronger training signal for physically verifiable wireless math**, enabling exploration of correct reasoning paths that token-level supervision does not reliably capture.
> >
> > We have incorporated this ablation and discussion into the revised manuscript.

---

> > > ### Author Response · Authors · 2025-11-24
> > >
> > > > W4: There are several writing issues and typos.
> > > >
> > >
> > > Thank you for pointing this out. We agree that clarity and polish are essential, especially for a Dataset Track paper. In the revision, we conducted a detailed proofread and corrected the reported typos, grammatical issues, and inconsistent terminology/notation. We also tightened several paragraphs for readability, clarified ambiguous phrasing (notably in the Introduction and Conclusion). We hope these edits address your concerns and make the presentation substantially cleaner.
> > >
> > > > Q1: During the paper filtering process, how is the relevance score assigned to each paper? Is this score automatically generated by the LLM or explicitly calculated based on predefined metrics?
> > > >
> > >
> > > Thank you for the question. The relevance score is **explicitly computed using predefined, non-LLM metrics**, and is fully reproducible. Concretely, for every crawled arXiv paper we calculate a **Relevance Score in [0,1]** as a weighted sum of keyword hits in the **title (weight 0.6)** and **abstract (weight 0.3)**, plus a **category bonus** reflecting core wireless venues (e.g., eess.SP: +0.4, [cs.NI](http://cs.ni/): +0.35, [cs.IT](http://cs.it/): +0.3). This scoring is implemented as a deterministic script and does not involve any model-generated judgment.
> > >
> > > Using this relevance score, we first perform a **keyword/category-based pre-filter** over ~47,000 papers to retain candidates that are topically aligned with wireless communications. Only **after** this deterministic step, we apply GPT-4o as a **second-stage filter** to identify papers with substantial mathematical density, yielding 3,186 papers before final selection.
> > >
> > > > Q2: Given that the human effort is still required to review each question, what advantages does the LLM-based automated evaluation provide? Moreover, it is encouraged to develop a fully end-to-end, automated evaluation framework without human intervention.
> > > >
> > >
> > > Thank you for this forward-looking suggestion. We agree that a fully automated pipeline is the ultimate goal, but we believe the current "Human-in-the-Loop" approach is strictly necessary for a dataset of this caliber.
> > >
> > > **1. The Necessity of PhD-Level Experts:**
> > > Unlike general math benchmarks (e.g., GSM8K), `WirelessMathBench-XL` involves graduate-level problems extracted from state-of-the-art research. Correctly validating these problems requires recognizing implicit physical constraints (e.g., causality, power budgets) and complex matrix logic that even GPT-4o occasionally misses. Our 6 human experts are **PhD students and postdocs** specializing in these subfields. Their review was critical: they rejected **~22%** of the LLM-generated questions that had passed the initial automated filter, ensuring the final benchmark is reliable.
> > >
> > > **2. The Advantage of LLM Automation (Scalability):**
> > > The LLM acts as a massive, efficient pre-filter. It processed an initial crawl of **~47,000 papers** to identify the top ~1,000 with high-quality mathematical content, a scale physically impossible for humans to review manually. This "funnel" approach allows our experts to focus their limited time only on high-potential candidates.
> > >
> > > **3. Path to Full Automation (Future Work):**
> > > We agree with your vision for an end-to-end framework. Paradoxically, high-quality human-verified datasets like ours are the prerequisite for building such systems. We added a discussion to the **Limitations and Future Work** section, explicitly stating that developing a robust "Auto-Judge" trained on `WirelessMathBench-XL` to replace human verification is a key direction for future research.

---

> > > > ### Author Response · Authors · 2025-11-24
> > > >
> > > > > Q3: Compared to general mathematical reasoning problems, what are the main challenges in training LLMs on wireless mathematical problems?
> > > > >
> > > >
> > > > Thank you for this insightful question. Compared to general mathematical reasoning (e.g., GSM8K, MATH), wireless mathematical problems introduce challenges that require grounding in physical systems rather than purely symbolic manipulation:
> > > >
> > > > 1. **Implicit physical constraints and validity.** Many constraints in wireless are *unstated but mandatory* (e.g., transmit power non-negativity (P \ge 0), causality in filtering, phase bounds ([0,2\pi)), feasibility of SINR/beamforming constraints). A derivation can be algebraically consistent yet physically invalid if these laws are ignored, which is a common failure mode for general LLMs.
> > > > 2. **High-dimensional complex-valued matrix calculus.** Wireless formulations rely heavily on complex matrices and non-commutative operators (e.g., Hermitian transpose (\mathbf{H}^H), trace, Kronecker products, block matrices in (\mathbb{C}^{N\times M})). Models must maintain dimension/operation consistency across multi-step derivations, which differs from the mostly real-valued, lower-dimensional algebra in standard math benchmarks.
> > > > 3. **Cross-disciplinary synthesis of mathematical tools.** A single wireless problem often requires choosing and combining the correct physical and mathematical framework (information theory bounds, convex/monotone optimization, stochastic geometry, signal processing identities). Success depends on selecting the right principle *before* computing, not just executing a known algebraic routine.
> > > >
> > > > We revised the Introduction to foreground these three points, clarifying why general math models underperform in wireless settings and motivating the need for a training-scale, physically verifiable benchmark and a constraint-aware post-training recipe.

---

### Author Response · Authors · 2025-11-24
**General Response to All Reviewers**

We sincerely thank Reviewers **GMo4, uN8k, qHbW, and UFY2** for the careful reading and constructive feedback. We are encouraged that all reviewers recognize the **community need and practical value** of a training-scale, expert-level benchmark for wireless technical mathematics. In particular, reviewers independently highlight *WirelessMathBench-XL* as filling a clear gap between evaluation-only datasets and training-ready resources, noting its breadth of 970 papers, realistic derivations, and dual-layer QA/curation (**GMo4, uN8k, qHbW, UFY2**).

Several reviewers further note that our **standardized comparisons across proprietary and open-source LLMs** make the benchmark useful for future research (**uN8k, qHbW**), and they find the **verification-only GRPO post-training recipe** appealing because it is simple, reproducible, and enabled by verifiable answers (**uN8k, qHbW, UFY2; also consistent with GMo4’s positive comments on our experimental validation**). Multiple reviewers also highlight the **surprising positive transfer to general mathematics** as an interesting scientific observation beyond a pure dataset contribution (**GMo4, qHbW, UFY2**).

To address the key concerns raised during the rebuttal period, we conducted some **new experiments and analyses**:

1. **Matched SFT baseline.**
We trained a standard SFT baseline on **Qwen2.5-3B-Base** using the *identical* WirelessMathBench-XL training set and evaluated checkpoints over 6 epochs. SFT peaks at **20.37%** overall (Epoch 4) and shows overfitting/instability thereafter, whereas **GRPO reaches 25.12%**, exceeding peak SFT by **+4.75% absolute (+23% relative)** under the same data and base model. This directly supports our claim that **verification-based RL is more effective than token-level supervision for physically-verifiable technical math**.
2. **GRPO on a Qwen3 model.**
We ran the identical GRPO pipeline on **Qwen3-4B-Base**. GRPO again yields clear gains (**13.63% → 17.87%, +4.24 absolute**) across MCQ / Fill-in / FEC, confirming that the method **generalizes across base model families** and is not tied to Qwen2.5.
3. **General-purpose benchmark evaluation.**
In addition to the out-of-domain math transfer already reported in the paper, we evaluated models on **MMLU, GPQA, IFEval, and HumanEval** before and after GRPO. We observe **no regression and consistent improvements** for both 3B and 7B models (notably on knowledge- and reasoning-heavy benchmarks). This provides stronger evidence that **domain-RL strengthens foundational reasoning rather than causing catastrophic forgetting**.
4. **Bias analysis of evaluation and subdomain behaviour.**
We conducted **two robustness checks** to assess whether our evaluation pipeline or subdomain composition could bias the reported results. First, our hierarchical verifier relies on **deterministic matching for the majority of test items**, while LLM-based semantic judging is used only on a **model-dependent subset (approximately 23%–37% across top models)**; cross-evaluator re-grading shows **10–20 percentage-point variance** on this subset, yielding a conservative worst-case overall impact of **about 4.6%–7.3% absolute**, with **rankings unchanged**. We also provide a problem-level **20-subdomain breakdown** on the 800-problem test set, showing GRPO improves **19/20** categories, including both large and small subfields, mitigating overfitting concerns.

In parallel, we revised the manuscript for clarity and positioning:

- We **rewrote the Introduction** to foreground the scientific findings enabled by the benchmark (verification-based RL > supervision for verifiable technical math; training-scale wireless corpus enables efficient specialization; positive transfer contradicts standard forgetting intuition).
- We incorporate the new SFT/Qwen3/general-benchmark/subdomain results into the main paper and appendix.
- We corrected writing issues and typos noted by reviewers.

Below, we respond to each reviewer point-by-point.

---

### Note · Authors · 2026-01-06

I have read and agree with the venue's withdrawal policy on behalf of myself and my co-authors.